# HUMAN-IN-THE-LOOP ADAPTIVE OPTIMIZATION FOR IMPROVED TIME SERIES FORECASTING

## ABSTRACT

Time-series forecasting models often produce *systematic* and predictable errors, even in critical domains such as energy, finance, and healthcare. We introduce a novel *post-training adaptive optimization* framework that improves forecast accuracy without retraining or architectural changes. Our approach adds a lightweight model-agnostic correction layer that automatically finds expressive output transformations optimized by reinforcement learning, contextual bandits, or genetic algorithms. Theoretically, we prove the benefit of an affine correction and quantify the expected performance gain together with its computational cost. The framework also supports an *optional* human-in-the-loop component: domain experts can guide corrections using natural language, which is parsed into actions by a language model. Across multiple benchmarks (e.g. electricity, weather, traffic), we observe consistent accuracy gains with minimal computational overhead. Our interactive demo (link) showcases the usability of the framework in real time. By combining automated post-hoc refinement with domain-expert corrections to the base forecasting model, our approach offers a lightweight yet powerful direction for practical forecasting systems.

## 1 INTRODUCTION

Time series forecasting is critical in domains such as finance (Krollner et al., 2010), healthcare (Kaushik et al., 2020), and energy management (Palma et al., 2024), where accurate predictions drive high-stakes decisions. Although modern machine learning models have improved forecasting performance, they still face two persistent limitations: (1) insufficient model expressiveness to capture complex, real-world patterns, and (2) difficulty incorporating domain expertise into predictions. Traditional forecast pipelines (Meisenbacher et al., 2022) often rely on rigid architectures and static assumptions, leading to systematic errors that domain experts can easily identify.

However, integrating expert feedback remains challenging: manual corrections are time consuming, and existing methods (Geweke & Whiteman, 2006; Girard et al., 2002) require extensive re-engineering or ensembling techniques (Khashei & Bijari, 2012). These limitations prevent models from adapting effectively to changing environments.

To address these issues, we propose a flexible, lightweight *post-training optimization* framework that improves forecasts without re-training the model. Our preliminary theoretical insight suggests the opportunity for such a post hoc correction. Building on this, we extend post-training correction into a broader optimization framework that adaptively adjusts model outputs using approaches such as reinforcement learning, bandits, or genetic algorithms.

Our proposed approach, illustrated in Figure 1 and Figure 2, is scalable, model-agnostic and accessible through an interactive web interface, making it practical for both researchers and practitioners.

Our novel approach introduces key features that distinguish it from previous work:

1. *Adaptive Model Augmentation*: Automatically identifies and applies expressive transformations that improve the performance of the forecast, expanding the model function class without architectural changes.

2. *Human-in-the-Loop (HITL)*: Optionally incorporates expert feedback, expressed in natural language and safely translated into a post-training action code, which is further optimized

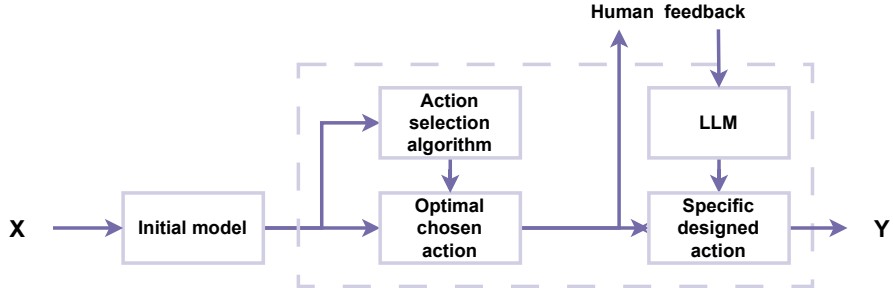

Figure 1: Overview of the forecasting pipeline: the initial model generates predictions from input $X$, which are refined by an action selection mechanism and optionally adjusted using human feedback interpreted by a language model (LLM), yielding the final output $Y$.

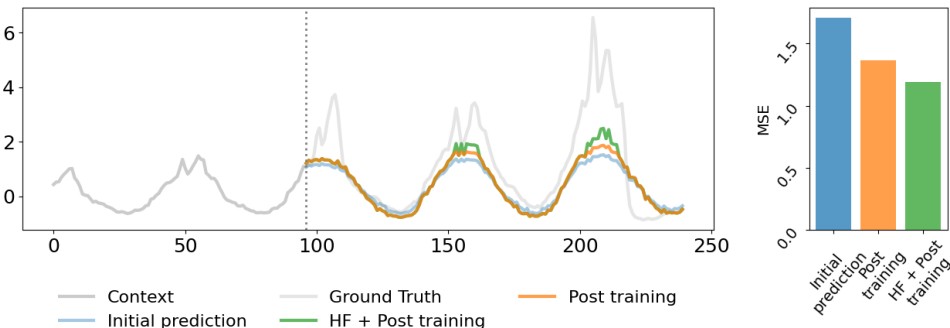

Figure 2: **(Left)** Ground truth, initial prediction, and corrected predictions produced by the proposed forecasting pipeline. The human feedback applied was: *"Increase values above a chosen quantile by 10% to 50%."*. **(Right)** Performance comparison of the different predictions: the base forecast, the automatically corrected output, and the correction incorporating human feedback.

via reinforcement learning, bandit methods, or genetic algorithms to iteratively improve and refine forecasts.

## 2 RELATED WORK

**Time Series Forecasting Models**     Time series forecasting has long been a fundamental task in statistical modeling. Traditional models such as *ARIMA* (Newbold, 1983), *SARIMA* (Korstanje, 2021), and *ETS* (Gardner Jr, 1985) work well for simple linear dynamics, but struggle with non-stationary or highly non-linear signals. Modern deep learning models, including *LSTMs* (Graves & Graves, 2012; Lin et al., 2023) and *Transformers* (Liu et al., 2023; Ilbert et al., 2024; Wu et al., 2021; Nie et al., 2023), offer improved expressiveness by learning long-range dependencies. Recent zero-shot models such as *TimesFM* (Das et al., 2024), *Chronos* (Ansari et al., 2024), and *Lag-LLaMA* (Rasul et al., 2023) further generalize across tasks via foundation model scaling. Despite these advances, existing models often exhibit systematic forecast errors and lack mechanisms to incorporate expert corrections.

Our work complements existing models by adding a post-training optimization layer that enhances performance without retraining and is compatible with any forecasting architecture.

**Post-Training and Human Feedback in Forecasting**     Incorporating expert knowledge into forecasting has a long history, from manual tuning and domain-specific feature engineering (Tavenard et al., 2020; Zhou, 2020; Verkade et al., 2013; Madadgar et al., 2014) to judgmental forecasting methods (Armstrong, 1986; Bunn & Wright, 1991; Webby & O'Connor, 1996). However, such methods are typically manual, hard to scale, and not integrated into learning pipelines. Recent work

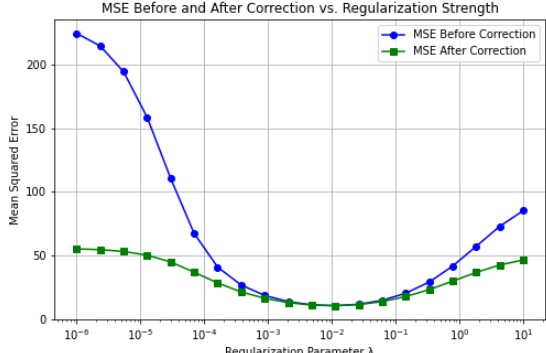

Figure 3: Illustration of the effect of affine post-training correction on ridge regression forecasts. The model is trained on a synthetic linear target. Results shown for 100 samples, 100 validation points, and 10,000 test points.

on human-in-the-loop learning - primarily in NLP (Liu et al., 2024a) - has explored expert-guided model refinement. In time series, systems such as *DelphAI* (Kupferschmidt et al., 2022) allow manual modification of model outputs and (Arvan et al., 2019) provide a comprehensive review of human input in forecasting.

Our approach advances this line of work in two key ways. First, it enables automatic post-training corrections via adaptive optimization using reinforcement learning, bandits, or genetic algorithms. Secondly, it optionally incorporates expert feedback through natural language, automatically translated into optimization actions by a large-language model (LLM). Unlike methods such as *TimeHF* (Qi et al., 2025), which require fine-tuning large models, our solution is model-agnostic, efficient and applies corrections at inference time.

## 3 METHODOLOGY

We propose a framework to improve time series forecasts through *post-training optimization*. It operates on any forecasting model, applying lightweight corrections without retraining. The system combines two components: (1) **automated prediction augmentation via dynamic optimization**, and (2) **optional human-in-the-loop feedback**. We first motivate the theoretical foundation and then describe the full pipeline.

### 3.1 THEORETICAL MOTIVATION FOR POST-TRAINING CORRECTION

Forecasting models often display systematic biases. These can be mitigated after training by applying affine transformations to outputs, leaving model parameters unchanged. For predictions $Y_{\text{pred}}$, the corrected forecast is

$$Y_{\text{corrected}} = a^* Y_{\text{pred}} + b^*,$$

with optimal parameters from validation statistics:

$$a^* = \frac{\text{Cov}(Y_{\text{true}}, Y_{\text{pred}})}{\text{Var}(Y_{\text{pred}})}, \quad b^* = \mathbb{E}[Y_{\text{true}}] - a^* \mathbb{E}[Y_{\text{pred}}].$$

**Theorem 1 (Affine Correction Reduces MSE)** *The above correction guaranties a lower or equal mean squared error (MSE):*

$$R_{before} - R_{after} = \left( \sqrt{Var(Y_{pred})} - \frac{Cov(Y_{true}, Y_{pred})}{\sqrt{Var(Y_{pred})}} \right)^2 \geq 0.$$

This result holds under distributional alignment of validation and test sets. Figure 3 shows the mean squared error as a function of the ridge regularization parameter for a ridge regression model with an added linear post-layer correction, illustrating the result of Theorem 1. Although affine post-hoc adjustments are effective, multi-step forecasts often require richer, dynamic corrections optimized via reinforcement learning, contextual bandits, or genetic algorithms (see Section 3.3).

## 3.2 Forecasting Model Setup

The framework is model-agnostic: it applies to classical (e.g., ARIMA, Prophet), deep learning (e.g., LSTM, Transformer), and foundation models (e.g., TimesFM, Chronos). Given a multivariate time series, of length $T$, $\{\mathbf{x}_1, \ldots, \mathbf{x}_T\}$ with $\mathbf{x}_t \in \mathbb{R}^d$, the base model outputs $\{\hat{\mathbf{y}}_{T+1}, \ldots, \hat{\mathbf{y}}_{T+H}\}$, which are refined post-training. Our objective is to design a post-training method that takes the output of the base model and produces corrected predictions, improving accuracy and generalization.

## 3.3 Post-Training Optimization via Action Space

To refine model predictions, we define a set of post-training transformations, or *actions*, each parameterized continuously. These actions are dynamically selected and tuned to minimize validation error.

- **Scale Amplitude:** Multiplies the full prediction series.
- **Piecewise Scaling:** Modifies high or low quantiles selectively.
- **Linear Trend:** Adds a slope or intercept term.
- **Min/Max Adjustment:** Boosts extrema to match observed dynamics.

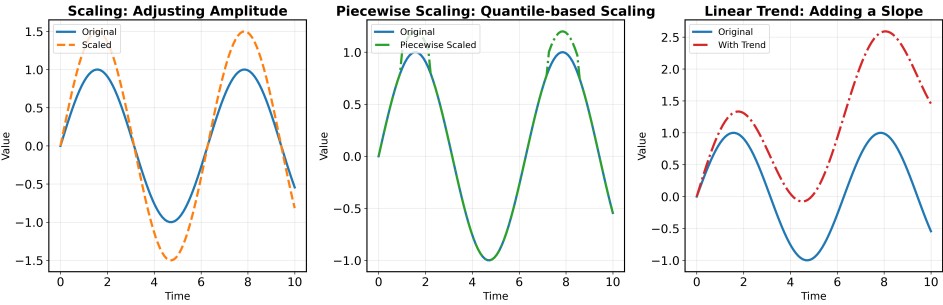

Figure 4: Examples of learned post-training actions. Each transformation operates on the model's forecast to reduce prediction error. Full action definitions are in Appendix A.1.

These interpretable actions form a flexible augmentation layer. They can be optimized efficiently and extended to task-specific needs, as discussed in later sections.

### 3.3.1 Optimizing Actions and Parameters

We frame the post-training refinement process as a joint optimization over a discrete set of actions and their associated continuous parameters. Discrete actions (e.g., scaling, shifting, trend) define transformation types, while parameters control their magnitude. The goal is to select and tune the best combination to minimize validation loss.

---

**Algorithm 1:** Forecast Augmentation via Post-Training Optimization

**Input:** Forecasting model $\mathcal{M}$, action space $\mathcal{A}$, validation data $D$
**Output:** Augmented model $\mathcal{M}_{\text{opt}}$, refined prediction $\hat{\mathbf{y}}$
1 Generate base predictions $\hat{\mathbf{y}} = \mathcal{M}(D)$
2 Define the loss function $\bar{\mathcal{L}}$ (e.g., MSE) on validation set
3 **for** *each iteration* **do**
4     Select candidate action(s) from $\mathcal{A}$
5     Optimize associated parameters (e.g., line search)
6     Apply transformation(s) to $\hat{\mathbf{y}}$
7     Evaluate $\bar{\mathcal{L}}$ and update strategy
8 Return best transformation sequence

---

### 3.3.2 DYNAMIC OPTIMIZATION STRATEGIES

We explore several strategies to solve this search problem efficiently:

- **Random search**: For each discrete action, we randomly generate several sets of continuous parameters. We then evaluate these and keep the set that gives the best performance for that action.

- **Bandit Algorithms (SH-HPO):** (Karnin et al., 2013) Each action is a bandit arm; its parameters are optimized independently (e.g., via line search). UCB balances exploration and exploitation to select the most rewarding transformations.

- **Reinforcement Learning (PPO):** (Schulman et al., 2017) Discretizing the parameter space allows us to train an RL agent that sequentially selects actions to minimize residual error. We use Proximal Policy Optimization (PPO) for stability.

- **Genetic Algorithms (GA):** (Holland, 1975) GA evolves action-parameter pairs through mutation and crossover, well-suited for large, multimodal spaces where gradients are unavailable or unreliable.

These techniques provide trade-offs between exploration depth and runtime. Our framework supports all three and can switch strategies based on task complexity.

### 3.3.3 WHY DISCRETE ACTIONS + CONTINUOUS PARAMETERS

The hybrid search space balances flexibility, efficiency, and interpretability without altering the base model. Advantages include: (i) reduced search complexity, (ii) human-readable corrections, (iii) faster convergence, and (iv) extensibility to new domains. To mitigate overfitting, we first evaluate actions on a validation set and then verify that they also improve performance on the training set. This "consistency check" helps ensure that selected actions yield genuine, generalizable gains rather than overfitting to the validation set. Empirically, we find that overfitting is rare in our experiments: cross-metric experiments in Appendix A.5 show strong agreement between training, validation, and test landscapes across different metrics throughout the episodes.

### 3.4 OPTIMIZATION STRATEGY: EMPIRICAL COMPARISON

We compare Random Search, SH-HPO (Successive Halving with UCB), Proximal Policy Optimization (PPO), and Genetic Algorithms (GA) on the ETTH1 dataset. All methods improve over the baseline, with SH-HPO giving the most consistent gains across horizons. Random Search also performs strongly, often matching SH-HPO while being simpler and more efficient. Their advantage likely stems from naturally handling discrete–continuous search spaces, whereas PPO and GA require full discretization, which increases complexity and introduces approximation errors that reduce effectiveness. Nevertheless, PPO and GA remain valuable for tasks needing long-term optimization, since they explore trajectories rather than making greedy, step-wise decisions. Despite lower performance here, they may be better suited to structured or sequential problems. For the remainder of our experiments, we adopt Random Search for its simplicity, efficiency, and competitive results. Full results and additional dataset comparisons are in the Appendix.

**Evaluation Metric for Post-Training.** To assess post-training effectiveness, we use the relative decrease in mean squared error (MSE). Let $MSE_{\text{before}}$ and $MSE_{\text{after}}$ denote the model's MSE before and after post-training. The relative improvement is

$$\mathcal{M} = \frac{MSE_{\text{before}} - MSE_{\text{after}}}{MSE_{\text{before}}}. \tag{1}$$

Positive $\mathcal{M}$ indicates reduced error, higher values reflect greater improvement, and negative $\mathcal{M}$ indicates post-training degraded performance. This normalized metric enables fair comparison across models and datasets with different MSE scales. Complementary experiments using other metrics are reported in Appendix A.5.

Table 1: Performance comparison of different optimization techniques. Reported values are percentage improvements in mean squared error (MSE) relative to baseline models, averaged over 10 trials on the **Nature** dataset. Standard deviations are shown as uncertainty.

| Model | Random | SH-HPO | RL (PPO) | GA |
|---|---|---|---|---|
| AutoFormer | $17.19\% \pm 5.3\%$ | $19.34\% \pm 5.1\%$ | $3.35\% \pm 1.2\%$ | $6.09\% \pm 1.5\%$ |
| Crossformer | $3.30\% \pm 2.1\%$ | $2.78\% \pm 1.9\%$ | $1.11\% \pm 1.4\%$ | $1.52\% \pm 1.3\%$ |
| DLinear | $1.96\% \pm 0.8\%$ | $1.57\% \pm 0.7\%$ | $2.20\% \pm 1.1\%$ | $1.61\% \pm 0.9\%$ |
| PatchTST | $-1.33\% \pm 1.2\%$ | $-0.81\% \pm 1.0\%$ | $0.26\% \pm 0.5\%$ | $0.29\% \pm 0.4\%$ |
| SegRNN | $1.70\% \pm 0.6\%$ | $2.59\% \pm 0.8\%$ | $0.61\% \pm 0.4\%$ | $0.59\% \pm 0.5\%$ |
| iTransformer | $3.10\% \pm 1.1\%$ | $3.85\% \pm 1.3\%$ | $1.33\% \pm 0.7\%$ | $1.50\% \pm 0.8\%$ |
| TimesFM | $4.94\% \pm 2.3\%$ | $5.43\% \pm 2.1\%$ | $3.48\% \pm 1.5\%$ | $2.62\% \pm 1.4\%$ |
| **Average** | **$4.84\% \pm 1.9\%$** | **$4.96\% \pm 1.9\%$** | **$1.76\% \pm 1.1\%$** | **$2.32\% \pm 1.3\%$** |

## 4 THEORETICAL ANALYSIS OF OUR BANDIT-BASED CORRECTION

Our method evaluates a large set of corrective actions (Section 3.3) and selects the best one using several candidate selection strategies. Among them, the bandit-based approach with the *Successive Halving* algorithm consistently achieves the strongest results in our experiments.

Natural questions are: *how quickly does this algorithm identify the best correction?* and *How does the validation budget affect its performance?* This section answers those questions theoretically. We focus on the simplest non-trivial setting of two corrective actions to present the key result; the general case ($K > 2$ actions) and all proofs are deferred to Appendix A.

**Why Successive Halving.** Successive Halving is a near-optimal best-arm identification algorithm (Karnin et al., 2013). It allocates more evaluations to promising actions while discarding the worse ones. This matches our setting, where evaluating each correction on the validation set is costly, and explains its superior empirical performance compared to uniform allocation.

**Corollary 1 (Convergence to the Best Correction)** *Consider two corrective actions $g_{1,\beta^*}$ and $g_{2,\beta^*}$ with $R(g_{1,\beta^*} \circ f_\theta) < R(g_{2,\beta^*} \circ f_\theta)$ and a validation budget of $T$ evaluations. Under Assumption 1 (Appendix A.11), the correction selected by our bandit-based procedure satisfies:*

$$\mathbb{E}\big[R(g_{k_T,\beta^*} \circ f_\theta)\big] \;\leq\; 2R(g_{1,\beta^*} \circ f_\theta) + 2\Delta\,\Phi\big(-\Delta\sqrt{T}\,\big), \quad \Delta = R(g_{2,\beta^*} \circ f_\theta) - R(g_{1,\beta^*} \circ f_\theta),$$

*where $\Phi$ is the standard Gaussian CDF.*

This bound shows that the expected risk of the selected correction converges exponentially fast in $\sqrt{T}$ to the risk of the best correction. Larger risk gaps $\Delta$ (i.e. more distinct corrective actions) lead to faster convergence.

**Corollary 2 (Budget to Outperform the Base Model)** *To guarantee that the selected correction improves on $f_\theta$, it suffices to allocate $T$ evaluations, with*

$$T \;\gtrsim\; \frac{4}{\Delta^2}\Big[\Phi^{-1}\Big(\frac{R(f_\theta) - 2R(g_{1,\beta^*} \circ f_\theta)}{2\Delta}\Big)\Big]^2 .$$

These results directly explain our empirical findings (Table 6): with a reasonable validation budget and sufficiently distinct corrective actions, the Successive Halving algorithm quickly identifies the best correction and improves forecasting accuracy. The general case and all proofs appear in Appendix A.11.

## 5 HUMAN-IN-THE-LOOP FEEDBACK INTEGRATION

Our framework operates autonomously but supports optional human-in-the-loop (HITL) refinement. Domain experts can provide natural language suggestions (e.g., *"Increase values above the 80th percentile"*), which are translated into candidate actions via an LLM (e.g., `Qwen2-72B-32K`). Crucially, human feedback never directly modifies predictions; instead, it must pass through the same optimization and validation pipeline as automated actions, ensuring safety and performance.

**From Natural Language to Candidate Actions** User prompts are converted into executable Python code and added to the candidate pool only after validation (Algorithm 2, Figure 5). If the initial suggestion fails, users can refine their input iteratively.

---

**Algorithm 2:** Human-in-the-Loop Feedback Integration

**Input:** User prompt $p$

1 **while** *true* **do**
2     $a \leftarrow \text{LLM}(p)$
3     **if** *Test(a)* **then**
4        AddActionToPool($a$)
5        **break**
6     $p \leftarrow \text{RequestNewPrompt}()$

---

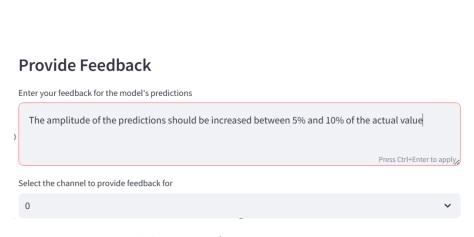

(a) User input prompt

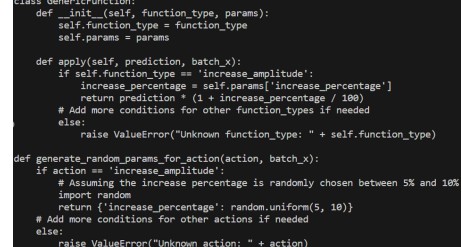

(b) Generated code for adaptive optimization

Figure 5: HITL pipeline: prompts are converted to code and validated before entering the candidate pool.

**Interactive Refinement and Safety** Users can iteratively refine prompts to improve candidate quality. All proposed actions undergo strict validation (*Test()*) for API compliance, execution, and forecast validity. The LLM never accesses raw time series data, preventing leakage or overfitting.

**Integration with Optimization** Validated human-proposed actions are evaluated alongside automated candidates in the optimization loop (Section 3.3). Only actions that improve performance are retained, ensuring robustness. Case studies in Section A.6.1 demonstrate this mechanism in practice.

## 6 EXPERIMENTS

We evaluate our framework across diverse real-world time series tasks, demonstrating consistent improvements in forecast accuracy using standard benchmarks and open-source implementations. All experiments were conducted on a server equipped with 2× Intel Xeon E5-2690 v4 CPUs (56 cores total), 512 GB RAM, and 6× NVIDIA Tesla P100 GPUs (16 GB each), though only one GPU was used per run.

### 6.1 SETUP

We evaluate our post-training optimization framework on energy consumption and OpenTS benchmark datasets (Zhou et al., 2021; Qiu et al., 2024) (details in Appendix A.2.1), across a wide range of forecasting models—from simple deep learning models such as DLinear (Zeng et al., 2023) to modern

architectures including SegRNN (Lin et al., 2023), iTransformer (Liu et al., 2023), PatchTST (Nie et al., 2023), Autoformer (Wu et al., 2021), Crossformer (Zhang & Yan, 2023), and Informer (Zhou et al., 2021). Forecast accuracy is reported using Mean Squared Error (MSE), averaged across multiple horizons (96, 192, 336, 720). Although performance is measured in MSE, our model-agnostic framework is compatible with any optimization or evaluation metric (e.g., MAE, MAPE, $R^2$); Section A.5 presents complementary experiments confirming its robustness and showing that it mitigates overfitting to the validation set.

While our framework can work with training samples alone, it generally requires for better performance a representative validation set for time series forecasting, in line with established practices. For benchmark datasets like `ETTh1`, `ETTh2`, `ETTm1`, and `ETTm2`, which provide an explicit validation set (designated by a *'val'* flag), we use these validation sets directly. For datasets without an explicit validation set, we apply the standard approach of temporally splitting the training data, allocating 30% to the validation set. This method aligns with common practices in the field of time series forecasting. We conduct a robustness analysis of the training-validation ratio in Section A.3.

### 6.2 RESULTS: ADAPTIVE OPTIMIZATION IMPROVES FORECASTING

Table 2 summarizes the impact of our post-training optimization. Across nearly all models and datasets, we observe significant MSE reductions with no retraining and minimal overhead (and **very few cases of overfitting in orange**). Rare cases of negative improvement can mostly be attributed to the stochastic nature of the search. The search algorithm iteratively evaluates the performance of actions on the validation set. As discussed in Subsection 3.3.3 and Appendix A.5, our approach is not prone to overfitting to the validation set; however, as shown in Section 4, it still induces a failure probability $\delta$.

#### 6.2.1 HUMAN-IN-THE-LOOP FEEDBACK

Human feedback can further enhance forecast accuracy by introducing domain knowledge not captured by the base model. Users provide natural language instructions (e.g., *"increase the amplitude of predictions below 0.5"*), which are converted into executable transformations via a large language model (LLM).

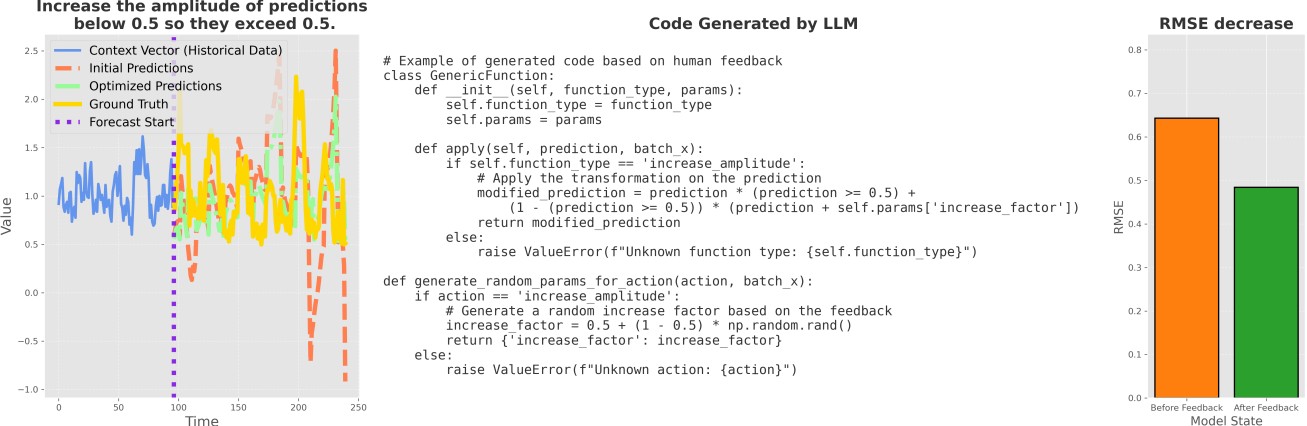

Figure 6: **(a)** Initial prediction vs. human-refined forecast. **(b)** Action code generated from natural language via LLM (`Qwen2-72B-32K`). **(c)** RMSE improvement post-feedback.

Figure 6 shows that integrating expert suggestions via HITL leads to tangible performance gains. Additional examples are presented in Appendix A.6 (see Figures 13, 14 and 15). The interface supports real-time experimentation, making human-guided optimization practical and intuitive. It is important to emphasize that the code generated by the LLM is guided by a strict template, which is detailed in the Appendix (Section A.6). This template constrains the possible code outputs, ensuring

| Methods | Autoformer | Crossformer | iTransformer | PatchTST | DLinear | SegRNN | Informer |
|---|---|---|---|---|---|---|---|
| ETTh1 | $0.61_{\pm 0.01} \to 0.51_{\pm 0.02}$ 
 (16.76%) | $0.54_{\pm 0.01} \to 0.52_{\pm 0.01}$ 
 (2.20%) | $0.45_{\pm 0.01} \to 0.44_{\pm 0.01}$ 
 (2.58%) | $0.46_{\pm 0.01} \to 0.47_{\pm 0.01}$ 
 (-2.25%) | $0.47_{\pm 0.01} \to 0.45_{\pm 0.01}$ 
 (1.38%) | $0.47_{\pm 0.01} \to 0.45_{\pm 0.01}$ 
 (1.31%) | $0.67_{\pm 0.01} \to 0.65_{\pm 0.01}$ 
 (3.00%) |
| ETTh2 | $0.65_{\pm 0.01} \to 0.55_{\pm 0.01}$ 
 (15.48%) | $0.60_{\pm 0.01} \to 0.57_{\pm 0.01}$ 
 (3.7%) | $0.60_{\pm 0.01} \to 0.57_{\pm 0.01}$ 
 (4.0%) | $0.60_{\pm 0.01} \to 0.57_{\pm 0.01}$ 
 (4.2%) | $0.60_{\pm 0.01} \to 0.57_{\pm 0.01}$ 
 (3.8%) | $0.60_{\pm 0.01} \to 0.57_{\pm 0.01}$ 
 (3.8%) | $0.60_{\pm 0.01} \to 0.57_{\pm 0.01}$ 
 (3.9%) |
| ETTm1 | $0.60_{\pm 0.01} \to 0.57_{\pm 0.01}$ 
 (7.37%) | $0.60_{\pm 0.01} \to 0.57_{\pm 0.01}$ 
 (3.7%) | $0.60_{\pm 0.01} \to 0.57_{\pm 0.01}$ 
 (4.0%) | $0.60_{\pm 0.01} \to 0.57_{\pm 0.01}$ 
 (4.2%) | $0.60_{\pm 0.01} \to 0.57_{\pm 0.01}$ 
 (3.8%) | $0.60_{\pm 0.01} \to 0.57_{\pm 0.01}$ 
 (3.8%) | $0.60_{\pm 0.01} \to 0.57_{\pm 0.01}$ 
 (3.9%) |
| ETTm2 | $0.60_{\pm 0.01} \to 0.57_{\pm 0.01}$ 
 (20.25%) | $3.82_{\pm 0.01} \to 3.81_{\pm 0.01}$ 
 (3.7%) | $0.60_{\pm 0.01} \to 0.57_{\pm 0.01}$ 
 (4.0%) | $0.60_{\pm 0.01} \to 0.57_{\pm 0.01}$ 
 (4.2%) | $0.60_{\pm 0.01} \to 0.57_{\pm 0.01}$ 
 (3.8%) | $0.60_{\pm 0.01} \to 0.57_{\pm 0.01}$ 
 (3.8%) | $0.60_{\pm 0.01} \to 0.57_{\pm 0.01}$ 
 (3.9%) |
| Dominick | $1.38_{\pm 0.01} \to 1.16_{\pm 0.01}$ 
 (15.42%) | $1.12_{\pm 0.01} \to 1.10_{\pm 0.01}$ 
 (1.00%) | $1.25_{\pm 0.01} \to 1.23_{\pm 0.01}$ 
 (0.89%) | $1.25_{\pm 0.01} \to 1.20_{\pm 0.01}$ 
 (3.83%) | $1.24_{\pm 0.01} \to 1.13_{\pm 0.01}$ 
 (7.97%) | $1.92_{\pm 0.01} \to 1.40_{\pm 0.01}$ 
 (27.21%) | $1.14_{\pm 0.01} \to 1.09_{\pm 0.01}$ 
 (4.28%) |
| Human | $0.40_{\pm 0.01} \to 0.33_{\pm 0.01}$ 
 (20.24%) | $0.30_{\pm 0.01} \to 0.25_{\pm 0.01}$ 
 (13.34%) | $0.60_{\pm 0.01} \to 0.15_{\pm 0.01}$ 
 (51.26%) | $0.62_{\pm 0.01} \to 0.15_{\pm 0.01}$ 
 (51.74%) | $0.96_{\pm 0.01} \to 0.26_{\pm 0.01}$ 
 (57.85%) | $0.60_{\pm 0.01} \to 0.25_{\pm 0.01}$ 
 (43.02%) | $0.30_{\pm 0.01} \to 0.21_{\pm 0.01}$ 
 (21.93%) |
| KDD | $1.25_{\pm 0.01} \to 0.95_{\pm 0.01}$ 
 (23.50%) | $0.80_{\pm 0.01} \to 0.80_{\pm 0.01}$ 
 (-0.03%) | $1.13_{\pm 0.01} \to 0.88_{\pm 0.01}$ 
 (21.09%) | $1.14_{\pm 0.01} \to 0.89_{\pm 0.01}$ 
 (21.83%) | $0.93_{\pm 0.01} \to 0.85_{\pm 0.01}$ 
 (8.54%) | $1.10_{\pm 0.01} \to 0.88_{\pm 0.01}$ 
 (19.14%) | $1.12_{\pm 0.01} \to 0.87_{\pm 0.01}$ 
 (22.15%) |
| Nature | $1.25_{\pm 0.01} \to 0.96_{\pm 0.01}$ 
 (21.69%) | $0.65_{\pm 0.01} \to 0.66_{\pm 0.01}$ 
 (-1.17%) | $0.34_{\pm 0.01} \to 0.33_{\pm 0.01}$ 
 (3.26%) | $0.26_{\pm 0.01} \to 0.25_{\pm 0.01}$ 
 (3.72%) | $0.95_{\pm 0.01} \to 0.91_{\pm 0.01}$ 
 (4.31%) | $1.02_{\pm 0.01} \to 0.94_{\pm 0.01}$ 
 (8.67%) | $0.91_{\pm 0.01} \to 0.90_{\pm 0.01}$ 
 (0.86%) |
| NASDAQ | $0.91_{\pm 0.01} \to 0.70_{\pm 0.01}$ 
 (22.39%) | $0.46_{\pm 0.01} \to 0.46_{\pm 0.01}$ 
 (-0.33%) | $0.80_{\pm 0.01} \to 0.63_{\pm 0.01}$ 
 (19.77%) | $0.79_{\pm 0.01} \to 0.68_{\pm 0.01}$ 
 (14.76%) | $0.74_{\pm 0.01} \to 0.68_{\pm 0.01}$ 
 (7.59%) | $0.78_{\pm 0.01} \to 0.62_{\pm 0.01}$ 
 (19.19%) | $0.85_{\pm 0.01} \to 0.79_{\pm 0.01}$ 
 (8.96%) |
| Pedestrian | $0.46_{\pm 0.01} \to 0.27_{\pm 0.01}$ 
 (40.01%) | $0.14_{\pm 0.01} \to 0.13_{\pm 0.01}$ 
 (0.26%) | $0.14_{\pm 0.01} \to 0.12_{\pm 0.01}$ 
 (9.43%) | $0.22_{\pm 0.01} \to 0.19_{\pm 0.01}$ 
 (9.54%) | $0.69_{\pm 0.01} \to 0.60_{\pm 0.01}$ 
 (14.20%) | $0.20_{\pm 0.01} \to 0.18_{\pm 0.01}$ 
 (6.94%) | $0.33_{\pm 0.01} \to 0.27_{\pm 0.01}$ 
 (15.87%) |
| Tourism | $0.24_{\pm 0.01} \to 0.22_{\pm 0.01}$ 
 (10.88%) | $0.16_{\pm 0.01} \to 0.14_{\pm 0.01}$ 
 (11.48%) | $0.31_{\pm 0.01} \to 0.14_{\pm 0.01}$ 
 (40.08%) | $0.31_{\pm 0.01} \to 0.14_{\pm 0.01}$ 
 (39.20%) | $0.52_{\pm 0.01} \to 0.25_{\pm 0.01}$ 
 (48.91%) | $0.24_{\pm 0.01} \to 0.12_{\pm 0.01}$ 
 (48.14%) | $0.22_{\pm 0.01} \to 0.18_{\pm 0.01}$ 
 (20.57%) |
| Vehicle trips | $1.39_{\pm 0.01} \to 1.13_{\pm 0.01}$ 
 (18.13%) | $0.83_{\pm 0.01} \to 0.82_{\pm 0.01}$ 
 (0.89%) | $1.02_{\pm 0.01} \to 0.84_{\pm 0.01}$ 
 (17.14%) | $0.98_{\pm 0.01} \to 0.80_{\pm 0.01}$ 
 (18.00%) | $1.35_{\pm 0.01} \to 1.15_{\pm 0.01}$ 
 (14.37%) | $1.68_{\pm 0.01} \to 1.05_{\pm 0.01}$ 
 (38.46%) | $1.21_{\pm 0.01} \to 0.95_{\pm 0.01}$ 
 (21.54%) |

Table 2: Mean squared error (MSE) ± standard deviation across multiple forecast horizons, before and after applying Adaptive Optimization ($\to$). Improvements are reported in teal when positive and orange when negative. The overall improvement across all models and datasets is 14.84%, with a peak improvement of 57.85%. This is based on 12 datasets and 7 time series models, with only 4 cases (out of 84) showing a performance decline, averaging -0.94% and a maximum of -2.25%.

that only valid and meaningful actions are generated. Additionally, the LLM not only generates the transformation code but also a function that creates a pool of candidate parameters. These parameters are then subject to optimization (as decribed in details in the automated optimization framework), with ineffective candidates being discarded if they do not improve the model's performance.

To assess the robustness of the framework, we provide several failure cases where the user's prompt is ambiguous or nonsensical. We investigate three cases in Figure 16, 17 and 18 with the prompt given in the title (e.g., `Optimize model to make it model like and being unoptimized`, `Replace everything by random noise` and `Taratata las palsma reality bonnegur selar`). In these cases, the system either discards the resulting action (if it does not lead to improvements) or the generated code fails to execute properly (improvement being 0%). These scenarios demonstrate the framework's ability to handle poor feedback and ensure that only actions leading to performance enhancement are retained.

The corresponding code generated by the LLM in these failure cases, along with additional examples, is provided in Appendix A.6.

**Remark 1 (On the safety of the human-in-the-loop framework)** *In addition to the automated optimization procedures, we provide an optional "safe mode". This mode displays the Python source code of the proposed actions for manual inspection by the user. The code is then automatically scanned using the malicious-source-code detector of Tsfaty & Fire (2023) before execution. This*

*detection step incurs only a small overhead, since each newly generated action is analyzed once at creation time, rather than at every inference step, and the corresponding code snippets are short.*

### 6.2.2 COMPUTATIONAL EFFICIENCY AND SCALABILITY

We evaluate optimization time across varying forecast horizons and action space sizes on the *ETTh1* dataset. Table 8 compares our post-training augmentation time against the minimum and maximum training times of standard forecasting models.

Table 3: Adaptive optimization time vs. base model training time (10 epochs).

| Horizon | 2 Actions | 4 Actions | 7 Actions | DLinear (min) | PatchTST (max) |
|---------|-----------|-----------|-----------|---------------|----------------|
| 96 | 3.2s ± 0.1 | 5.4s ± 0.2 | 12.0s ± 1.3 | 20.3s | 144.4s |
| 192 | 6.1s ± 0.3 | 9.7s ± 0.4 | 22.7s ± 1.5 | 22.3s | 146.2s |
| 336 | 12.7s ± 0.5 | 18.3s ± 0.6 | 30.1s ± 1.0 | 24.4s | 148.8s |
| 720 | 24.3s ± 1.1 | 35.2s ± 1.4 | 45.1s ± 1.8 | 27.3s | 151.8s |

Even for long horizons and expanded action spaces, our optimization time remains well below the training cost of most models, confirming the framework's suitability for real-time applications and large-scale deployment.

## 7 CONCLUSION

We presented a model-agnostic framework for time series forecasting that enhances predictions through post-training optimization and optional human-in-the-loop refinement. Unlike retraining-based methods, our approach applies lightweight, interpretable transformations, yielding consistent accuracy gains across diverse models and datasets at low cost. The framework is broadly compatible, fast, and interpretable, allowing for seamless integration of natural language feedback through LLMs. Its effectiveness depends on the quality of the base model, and LLM-based feedback translation may vary with prompt clarity; robustness to ambiguous instructions and access to a representative validation set remain open challenges.

Future work includes richer transformations (e.g. monotone, piecewise-affine, uncertainty-aware), stronger LLM alignment via structured prompting and automatic tests/guardrails, and applications to multimodal and streaming time series with online updates and drift handling.

## REPRODUCIBILITY STATEMENT

We detail the complete methodology and experimental settings in section 3 and section 6, with theoretical assumptions and results in section 4 and proofs in subsubsection A.10.3. The action space, search strategies, additional analyzes, datasets, preprocessing, splits, and model configurations appear in the main paper and are detailed in the beginning of the appendix. We provide an interactive demo to reproduce and test our approach. The hardware details for our runs are reported in section 6. To ease verification, we include: (i) fixed seeds, (ii) YAML configs for hyper-parameters, (iii) a single entry-point script for each experiment, and (iv) checks that validate data splits and metrics.

## LLM USAGE STATEMENT

We used large language models *solely for language editing* (grammar and clarity). An LLM did not generate technical claims, equations, algorithms, hyperparameters, or experimental decisions. All content was verified by the authors. We disclose this limited use here and in the submission form.

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

## A  APPENDIX

TABLE OF CONTENTS

ABSTRACT

This supplementary material provides an extended discussion and additional details supporting the main paper on *Human-in-the-Loop Adaptive Optimization for Improved Time Series Forecasting*. We first delve deeper into the mathematical formulation of the different actions used in our framework, offering visualizations to better illustrate their roles and impacts on model performance.

Next, we provide an in-depth exploration of the adaptive optimization algorithms employed within our approach, detailing their integration into state-of-the-art time series forecasting models. Supplementary experiments are included to showcase the effectiveness of our adaptive optimization -enhanced models across various datasets, comparing them against baseline methods to highlight performance improvements.

Human feedback is also a central aspect of our framework. In this section, we demonstrate how human feedback can be integrated into the post-training process through several real-world examples, illustrating the subjective nature of human input and its positive impact on model fine-tuning.

Finally, we offer detailed instructions on how to reproduce the experiments and results presented in this work. This includes guidance on using the provided code and graphical interface, enabling users to easily test and customize our framework for their own time series forecasting tasks. All code and resources are made publicly available for further exploration and use by the research community.

A.1    MATHEMATICAL DEFINITIONS AND VISUALIZATIONS OF THE POOL OF ACTIONS

Before presenting the mathematical definitions in the table, let's define the notation used in the transformations:

- $\mathbf{x}$: The original time series or predictions (before transformation), where each $x_t$ is the value at time $t$.
- $\mathbf{y}$: The transformed time series or predictions, resulting from applying one of the post-training actions.
- $x_t$: The value at time step $t$ in the original time series.
- $y_t$: The transformed value at time step $t$ in the new series.
- $x_{\max}$: The maximum value in the time series $\mathbf{x}$ across all time steps.
- $x_{\min}$: The minimum value in the time series $\mathbf{x}$ across all time steps.
- $\bar{x}$: The average value of the time series $\mathbf{x}$ over all time steps.
- $Q_\delta$: The $\delta$-th quantile of the values in $\mathbf{x}$, which corresponds to the value at the specified percentile of the distribution of $\mathbf{x}$.
- $\Delta$: The amount by which to shift the time series in the "Shift Series" action (in terms of time steps).
- $s$: The slope parameter for adding a linear trend to the time series, representing a change in the amplitude of the series over time.
- $b$: The intercept parameter for adding a linear trend to the time series, adjusting the average level of the series.
- $f$: The factor used in scaling operations, such as scaling the amplitude or adjusting the minimum/maximum values.
- $\sigma$: The standard deviation parameter for noise addition, influencing the spread of the generated noise.
- $t$: The time step index, which ranges from 1 to $H$, where $H$ is the total number of time steps (the horizon) in the time series.

The following table summarizes each action's mathematical operation and the continuous parameters involved, along with their respective ranges.

These post-training actions modify time series predictions through mathematical transformations targeting trends, amplitudes, and frequency/phase. Each transformation is defined mathematically, with adjustable parameters like scaling factors and thresholds to optimize performance.

| Action Name | Mathematical Definition | Continuous Parameters (Range) |
|---|---|---|
| **Trend Modifications** | | |
| *Linear Trend Slope* | $\mathbf{y} = \mathbf{x} + \left(\frac{s}{100} \cdot (x_{\max} - x_{\min})\right) \cdot t$ | $s \in (-5, 5), t \in [H]$ |
| *Linear Trend Intercept* | $\mathbf{y} = \mathbf{x} + \left(\frac{b}{100} \cdot \bar{x}\right)$ | $b \in (-5, 5)$ |
| **Piecewise Scaling** | | |
| *Piecewise Scale High* | $\mathbf{y} = \mathbf{x} \cdot \left(1 + \frac{f}{100}\right)$ for $x_t \leq Q_\delta$ | $\delta \in (70, 100), f \in (-1, 10)$ |
| *Piecewise Scale Low* | $\mathbf{y} = \mathbf{x} \cdot \left(1 + \frac{f}{100}\right)$ for $x_t > Q_\delta$ | $\delta \in (0, 30), f \in (-1, 10)$ |
| **Frequency and Phase** | | |
| *Swap Series* | $y_t = -(\mathbf{x} - \bar{x}) + \bar{x}$ | None |
| *Shift Series* | $y_t = x_{t+\Delta}$ | $\Delta \in (-200, 200)$ |
| **Amplitude Modifications** | | |
| *Scale Amplitude* | $\mathbf{y} = \mathbf{x} \cdot \left(1 + \frac{f}{100}\right)$ | $f \in (-5, 5)$ |
| *Add Noise* | $\mathbf{y} = \mathbf{x} + \epsilon$ where $\epsilon \sim \mathcal{N}(0, \frac{\sigma}{100} \cdot |x_t|)$ | $\sigma \in (10, 30)$ |
| *Increase Minimum Factor* | $\mathbf{y} = \mathbf{x} \cdot (1 + \frac{f}{100})$ for $x_t \leq Q_{10\%}$ | $f \in (-1, 10)$ |

Table 4: Mathematical definitions and continuous parameters for each post-training action. The range for each parameter is specified to guide the tuning of each transformation. Note: $x_t$ is the initial prediction and $y_t$ is the transformed prediction for each sample and each dimension. $x_{\max}$ and $x_{\min}$ are minimum and maximum values of $x_t$ over all horizons $H$ and $\bar{x}$ is the average value. $Q_\delta$ is the $\delta$-quantile of the vector $\mathbf{x}$.

Figure 7 shows the original time series alongside the transformed series, with each subplot illustrating a different post-training action applied.

## A.2 DETAILS ON EXPERIMENTAL SETUP: DATASETS AND MODELS

### A.2.1 DATASETS

In this section, we provide a summary of the datasets used in our analysis. The following table outlines the dataset names, their sources, key characteristics, and the corresponding references for the papers that describe each dataset.

Table 5: Datasets Overview

| Dataset Name | Source and Reference | Characteristics |
|---|---|---|
| ETTh1 | ETTh (Electricity) Benchmark Zhou et al. (2021) | 1-hour-level time-series with 6 features and "oil temperature" as the target. Train/val/test split: 12/4/4 months. |
| ETTh2 | ETTh (Electricity) Benchmark Zhou et al. (2021) | 1-hour-level time-series with 6 features and "oil temperature" as the target. Includes more features than ETTh1. |
| ETTm1 | ETTh (Electricity) Benchmark Zhou et al. (2021) | 15-minute-level time-series with 6 features and "oil temperature" as the target. Train/val/test split: 12/4/4 months. |

**Datasets Overview (Continued)**

| Dataset Name | Source and Reference | Characteristics |
|---|---|---|
| ETTm2 | ETTh (Electricity) Benchmark Zhou et al. (2021) | 15-minute-level time-series, similar to ETTm1, with different subsets for long-term forecasting. |
| Dominick | Open TS Benchmark Qiu et al. (2024) | 115704 weekly time series representing the profit of individual stock keeping units from a retailer. |
| Nature | Open TS Benchmark Qiu et al. (2024) | |
| Human | Open TS Benchmark Qiu et al. (2024) | Time-series data for human activity recognition, captured by wearable devices. |
| NASDAQ | Open TS Benchmark Qiu et al. (2024) | Stock market data from NASDAQ. Used for financial forecasting challenges. |
| KDD Cup | Open TS Benchmark Qiu et al. (2024) | |
| Pedestrian | Open TS Benchmark Qiu et al. (2024) | Pedestrian count data from urban settings, used for mobility prediction. |
| Tourism | Open TS Benchmark Qiu et al. (2024) | Tourism demand data, used for forecasting seasonal trends. |
| Vehicle Trips | Open TS Benchmark Qiu et al. (2024) | Vehicle trip data, used for urban mobility and traffic pattern forecasting. |

### A.2.2 TIME SERIES MODELS

In this work, we utilize several state-of-the-art time series forecasting models, all of which are part of the framework described in Liu et al. (2024b). These models are trained for 10 epochs, with early stopping applied on the validation set to prevent overfitting. Below, we briefly describe each of the models used:

- **Autoformer** Wu et al. (2021): A deep learning model designed to capture long-term dependencies and seasonality in time-series data by leveraging an attention mechanism.
- **Crossformer** Zhang & Yan (2023): This model integrates cross-attention mechanisms to effectively model both long-range and local dependencies in time-series forecasting.
- **PatchTST** Nie et al. (2023): A vision transformer-based model that divides time-series data into patches to capture temporal dependencies, providing superior performance in forecasting.
- **DLinear** Zeng et al. (2023): A linear decomposition model that separates the time series into trend and seasonal components for more interpretable and efficient forecasting.
- **Informer** Zhou et al. (2021): A transformer-based model that focuses on efficiency for long-term forecasting by using a self-attention mechanism and probabilistic forecast outputs.
- **SegRNN** Lin et al. (2023): A sequential deep learning model that combines segmentation with recurrent neural networks to handle irregular time-series data.

Each of these models has demonstrated strong performance in time series forecasting tasks, and we have used them to compare their abilities on the datasets described earlier.

### A.3 MORE EXPERIMENTS ON THE REINFORCEMENT AUTOMATED LOOP

#### A.3.1 ROBUSTNESS ANALYSIS WITH RESPECT TO VALIDATION SETS

In our framework, the choice of the validation set is quite important. For well-known benchmark datasets like ETTh1, ETTh2, ETTm1, and ETTm2, we use the provided validation sets as specified

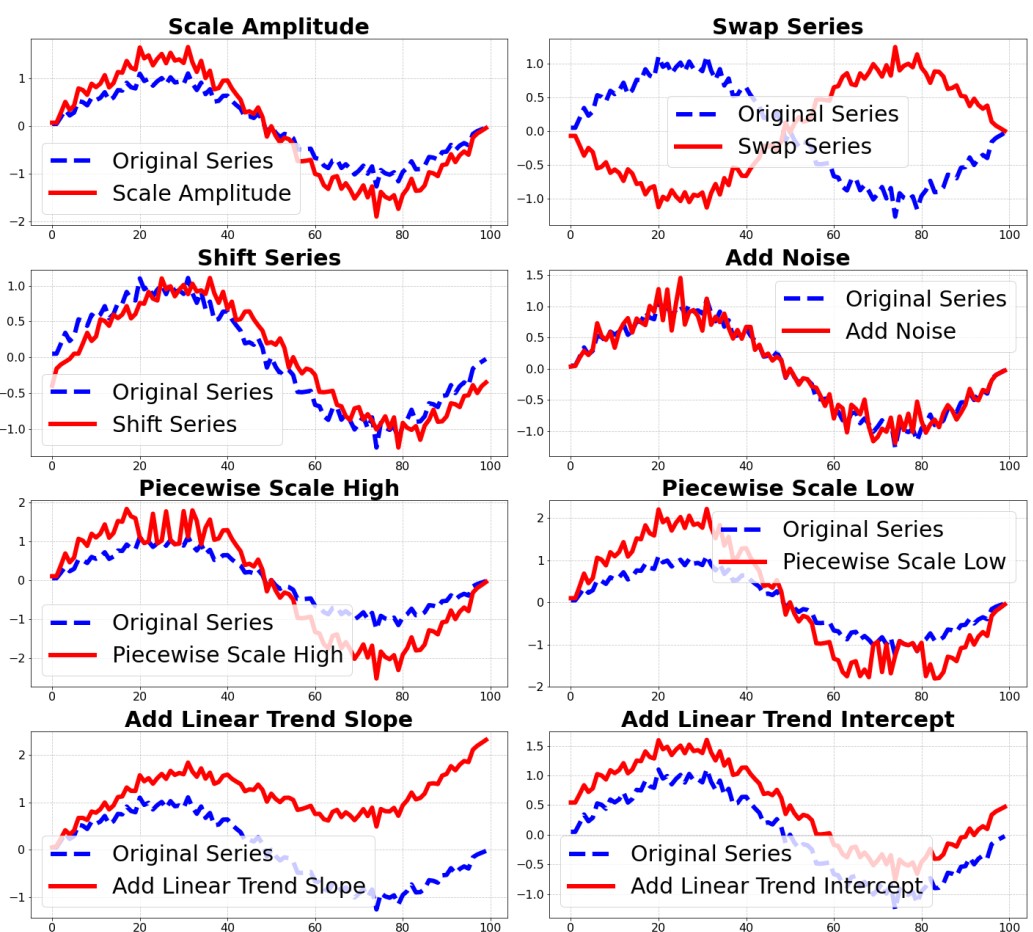

Figure 7: Visual representation of the time series transformations. Each subplot illustrates a different post-training action applied to the original time series.

in the dataset documentation. For datasets without a predefined validation set, we split the training data temporally, using 30% of the data for validation, in line with common practices.

To assess the robustness of our approach, we conduct additional experiments on the `ETTm1` dataset. While `ETTm1` provides an explicit validation set, we discard it for this analysis and perform our own temporal split. This allows us to investigate how model performance varies with different validation set sizes. We experiment with three models *DLinear*, *PatchTST*, and *SegRNN* and test multiple validation set sizes, from smaller to larger subsets. The results reveal a consistent improvement of approximately 10% as the validation set size increases, with a monotonic improvement over larger validation sets observed across all models.

Each experiment is repeated 5 times, with train-validation splits shuffled to ensure robustness. The standard deviation is shown as a shaded region around the mean performance, providing a clear view of variability.

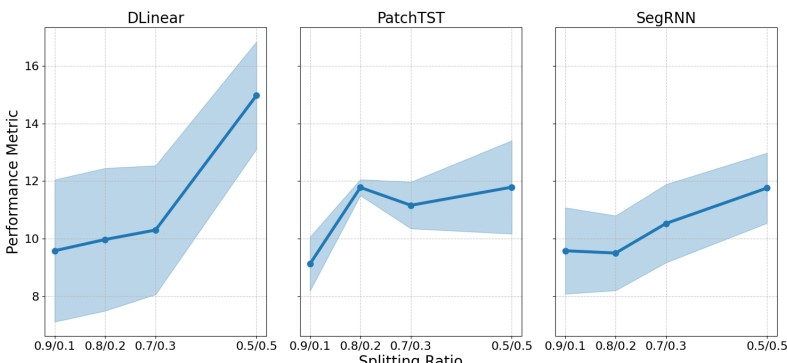

Figure 8: Quantitative MSE reduction (Performance metric) with respect to validation set size for DLinear, PatchTST, and SegRNN. We observe an MSE reduction of around 10% across different validation splits.

### A.3.2 QUANTITATIVE ANALYSIS OF THE OPTIMIZATION PROCESS

We present a qualitative analysis of the optimization process, illustrating the improvement in forecasting at different stages. The initial forecast is shown in red, representing the model's performance at the beginning of the optimization. The middle prediction, made after 5 episodes, is also shown to demonstrate the progress. Finally, the forecast after the optimization process is completed shows the model's final performance.

The model used for this prediction is *PatchTST*, and the dataset is `ETTm1`. In black, we highlight all the unsuccessful actions attempted during the optimization process.

From this analysis, we can clearly observe the improvement in forecast accuracy over time, driven by the optimization process, which refines the predictions based on a fixed set of actions.

### A.3.3 COMPARISON BETWEEN OPTIMIZATION STRATEGY OVER THE POOL OF ACTIONS

We compare the performance of four different classes of algorithms:

1. **Random search** where each discrete action is evaluated by randomly sampling continuous parameters and selecting the best-performing configuration.

2. **Bandit algorithm**, which considers each class of actions as an arm and optimizes using line search the parameter called `SH-HPO`.

3. **Reinforcement learning algorithm (PPO)**, which discretizes the set of actions and implements the PPO algorithm (Schulman et al., 2017) denoted `RL(PPO)`.

4. **Genetic algorithm (GA)** (Sampson, 1976), which discretizes the set of actions and performs a genetic algorithm denoted `GA`.

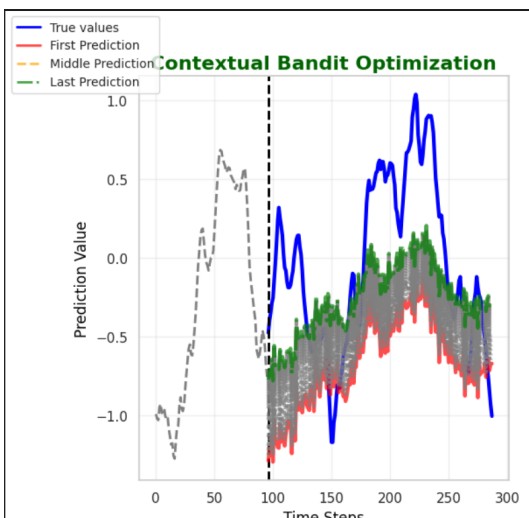

Figure 9: Qualitative improvement during the optimization process. The base learner is `PatchTST` on *ETTm1*. The validation set is the one explicitly provided by the benchmark. One sample from a specific channel is provided for illustration.

We present the result for several time series models and for five different datasets in Table 6.

The metric used to measure the efficiency of post-training is the relative decrease in Mean Squared Error (MSE) observed after post-training. Specifically, given the MSE before post-training, $MSE_{\text{before}}$, and the MSE after post-training, $MSE_{\text{after}}$, the relative decrease in MSE, $\mathcal{M}$, is calculated as:

$$\mathcal{M} = \frac{MSE_{\text{before}} - MSE_{\text{after}}}{MSE_{\text{before}}} \tag{2}$$

A positive value of $\mathcal{M}$ indicates that post-training has reduced the MSE, with larger positive values signifying greater improvement. Conversely, a negative value indicates degradation compared to the initial prediction.

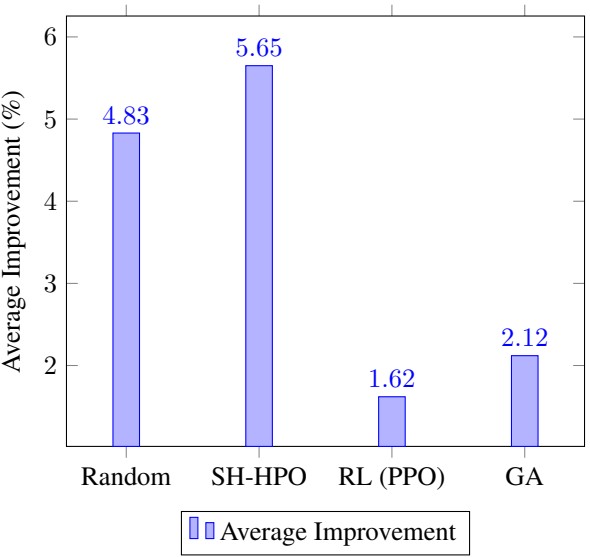

Figure 10: Average improvement across all models and datasets for each method.

| Models | Datasets | Random | SH-HPO | RL (PPO) | GA |
|---|---|---|---|---|---|
| **Autoformer** | ETTh1 (96) | 12.77% | 19.85% | 2.71% | 6.76% |
| | ETTh1 (192) | 18.42% | 17.32% | 3.14% | 6.20% |
| | ETTh1 (336) | 13.09% | 12.91% | 3.54% | 4.43% |
| | ETTh1 (792) | 24.48% | 27.26% | 4.02% | 6.96% |
| | **Average** | **17.19%** | **19.34%** | **3.35%** | **6.09%** |
| **Crossformer** | ETTh1 (96) | 5.49% | 4.01% | 2.71% | 0.16% |
| | ETTh1 (192) | 2.05% | 3.80% | 1.71% | 3.48% |
| | ETTh1 (336) | 3.13% | 3.13% | 0.14% | 1.37% |
| | ETTh1 (792) | 2.53% | 0.17% | -0.14% | 1.07% |
| | **Average** | **3.30%** | **2.78%** | **1.11%** | **1.52%** |
| **PatchTST** | ETTh1 (96) | -0.99% | 0.15% | 0.40% | 0.62% |
| | ETTh1 (192) | -0.06% | -1.13% | 0.12% | 0.23% |
| | ETTh1 (336) | -3.13% | 0.23% | 0.41% | 0.19% |
| | ETTh1 (772) | -1.12% | -2.50% | 0.12% | 0.14% |
| | **Average** | **-1.33%** | **-0.81%** | **0.26%** | **0.29%** |
| **SegRNN** | ETTh1 (96) | 0.80% | 1.22% | 0.13% | 0.06% |
| | ETTh1 (192) | 1.24% | 1.56% | 0.73% | 0.68% |
| | ETTh1 (336) | 2.38% | 3.76% | 0.71% | 0.36% |
| | ETTh1 (772) | 2.39% | 3.81% | 0.87% | 1.26% |
| | **Average** | **1.70%** | **2.59%** | **0.61%** | **0.59%** |
| **DLinear** | ETTh1 (96) | 1.50% | 1.40% | 0.97% | 1.24% |
| | ETTh1 (192) | 2.03% | 2.18% | 1.39% | 2.35% |
| | ETTh1 (336) | 2.96% | 5.07% | 4.62% | 3.96% |
| | ETTh1 (772) | 1.33% | -2.37% | 1.82% | -1.11% |
| | **Average** | **1.96%** | **1.57%** | **2.20%** | **1.61%** |
| **Informer** | ETTh1 (96) | 12.98% | 6.83% | 6.12% | 4.87% |
| | ETTh1 (192) | 8.89% | 7.28% | 3.74% | 2.17% |
| | ETTh1 (336) | 1.68% | 4.01% | 3.81% | 2.49% |
| | ETTh1 (772) | -3.80% | 3.61% | 0.26% | 0.94% |
| | **Average** | **4.94%** | **5.43%** | **3.48%** | **2.62%** |
| **iTransformer** | ETTh1 (96) | 2.16% | 4.83% | 0.41% | 1.05% |
| | ETTh1 (192) | 2.79% | 1.89% | 1.26% | 1.03% |
| | ETTh1 (336) | 3.22% | 4.01% | 1.32% | 1.78% |
| | ETTh1 (772) | 4.23% | 4.67% | 2.34% | 2.15% |
| | **Average** | **3.10%** | **3.85%** | **1.33%** | **1.50%** |
| **Overall Average** | | **4.83%** | **5.65%** | **1.62%** | **2.12%** |

Table 6: Results of applying different algorithms (Random, SH-HPO, RL (PPO), GA) to various time series forecasting models and datasets. Each cell shows the observed improvement for the respective algorithm, model, and dataset. The improvements are measured in percentage points.

### A.3.4 EXPERIMENTS FOR THE SH-HPO ON ALL DATASETS AND HORIZONS

In the table, for each dataset, we evaluate performance at different horizon lengths (96, 192, 336, and 720), which are shown in the first four rows. The last row for each dataset represents the average performance improvement across all horizon lengths.

| Methods | Autoformer | Crossformer | iTransformer | PatchTST | DLinear | SegRNN | Informer |
|---|---|---|---|---|---|---|---|
| ETTh1 | $0.52_{\pm0.04}\to\mathbf{0.45}_{\pm0.03}$ (12.94%) | $0.41_{\pm0.02}\to\mathbf{0.39}_{\pm0.01}$ (3.41%) | $0.41_{\pm0.02}\to\mathbf{0.40}_{\pm0.01}$ (1.97%) | $0.40_{\pm0.02}\to0.41_{\pm0.01}$ (-1.31%) | $0.41_{\pm0.02}\to\mathbf{0.40}_{\pm0.01}$ (1.47%) | $0.39_{\pm0.02}\to\mathbf{0.38}_{\pm0.01}$ (1.22%) | $0.57_{\pm0.02}\to\mathbf{0.56}_{\pm0.01}$ (1.55%) |
|  | $0.59_{\pm0.03}\to\mathbf{0.48}_{\pm0.02}$ (17.75%) | $0.48_{\pm0.02}\to\mathbf{0.46}_{\pm0.02}$ (3.43%) | $0.45_{\pm0.03}\to\mathbf{0.44}_{\pm0.02}$ (2.77%) | $0.44_{\pm0.03}\to0.44_{\pm0.02}$ (-0.94%) | $0.46_{\pm0.03}\to\mathbf{0.45}_{\pm0.02}$ (2.15%) | $0.43_{\pm0.03}\to\mathbf{0.42}_{\pm0.02}$ (1.58%) | $0.67_{\pm0.03}\to\mathbf{0.62}_{\pm0.02}$ (7.10%) |
|  | $0.65_{\pm0.02}\to\mathbf{0.57}_{\pm0.01}$ (11.51%) | $0.59_{\pm0.02}\to\mathbf{0.58}_{\pm0.01}$ (0.67%) | $0.49_{\pm0.02}\to\mathbf{0.47}_{\pm0.01}$ (3.54%) | $0.47_{\pm0.02}\to0.48_{\pm0.01}$ (-3.38%) | $0.50_{\pm0.02}\to\mathbf{0.48}_{\pm0.01}$ (2.58%) | $0.50_{\pm0.02}\to\mathbf{0.47}_{\pm0.01}$ (1.31%) | $0.70_{\pm0.02}\to\mathbf{0.68}_{\pm0.01}$ (2.56%) |
|  | $0.71_{\pm0.02}\to\mathbf{0.53}_{\pm0.03}$ (24.85%) | $0.70_{\pm0.02}\to\mathbf{0.68}_{\pm0.01}$ (2.76%) | $0.48_{\pm0.02}\to\mathbf{0.47}_{\pm0.01}$ (2.05%) | $0.55_{\pm0.02}\to0.56_{\pm0.01}$ (-3.37%) | $0.51_{\pm0.02}\to0.50_{\pm0.01}$ (-0.68%) | $0.55_{\pm0.02}\to\mathbf{0.52}_{\pm0.01}$ (1.04%) | $0.75_{\pm0.02}\to\mathbf{0.74}_{\pm0.01}$ (0.75%) |
|  | $0.61_{\pm0.01}\to\mathbf{0.51}_{\pm0.02}$ (16.76%) | $0.54_{\pm0.01}\to\mathbf{0.52}_{\pm0.01}$ (2.20%) | $0.45_{\pm0.01}\to\mathbf{0.44}_{\pm0.01}$ (2.58%) | $0.46_{\pm0.01}\to0.47_{\pm0.01}$ (-2.25%) | $0.47_{\pm0.01}\to\mathbf{0.45}_{\pm0.01}$ (1.38%) | $0.47_{\pm0.01}\to\mathbf{0.45}_{\pm0.01}$ (1.31%) | $0.67_{\pm0.01}\to\mathbf{0.65}_{\pm0.01}$ (3.00%) |
| ETTh2 | $0.55_{\pm0.02}\to\mathbf{0.44}_{\pm0.01}$ (20.87%) | $1.10_{\pm0.02}\to\mathbf{1.04}_{\pm0.01}$ (5.38%) | $0.33_{\pm0.02}\to\mathbf{0.32}_{\pm0.01}$ (2.45%) | $0.32_{\pm0.02}\to\mathbf{0.31}_{\pm0.01}$ (3.23%) | $0.38_{\pm0.02}\to\mathbf{0.33}_{\pm0.01}$ (15.12%) | $0.31_{\pm0.02}\to\mathbf{0.30}_{\pm0.01}$ (3.51%) | $0.38_{\pm0.02}\to\mathbf{0.37}_{\pm0.01}$ (1.89%) |
|  | $0.78_{\pm0.03}\to\mathbf{0.64}_{\pm0.02}$ (18.65%) | $1.59_{\pm0.03}\to\mathbf{1.58}_{\pm0.02}$ (0.58%) | $0.40_{\pm0.03}\to\mathbf{0.39}_{\pm0.02}$ (1.60%) | $0.40_{\pm0.03}\to\mathbf{0.39}_{\pm0.02}$ (1.87%) | $0.50_{\pm0.03}\to\mathbf{0.47}_{\pm0.02}$ (6.54%) | $0.39_{\pm0.03}\to\mathbf{0.38}_{\pm0.02}$ (-5.88%) | $0.51_{\pm0.03}\to\mathbf{0.47}_{\pm0.02}$ (7.13%) |
|  | $0.75_{\pm0.02}\to\mathbf{0.63}_{\pm0.01}$ (16.69%) | $6.57_{\pm0.02}\to\mathbf{6.57}_{\pm0.01}$ (0.00%) | $0.44_{\pm0.02}\to\mathbf{0.43}_{\pm0.01}$ (2.98%) | $0.43_{\pm0.02}\to0.44_{\pm0.01}$ (-2.40%) | $0.61_{\pm0.02}\to\mathbf{0.57}_{\pm0.01}$ (6.15%) | $0.50_{\pm0.02}\to\mathbf{0.47}_{\pm0.01}$ (3.61%) | $0.46_{\pm0.02}\to\mathbf{0.44}_{\pm0.01}$ (3.84%) |
|  | $0.55_{\pm0.02}\to\mathbf{0.52}_{\pm0.01}$ (5.69%) | $0.55_{\pm0.02}\to\mathbf{0.52}_{\pm0.01}$ (-0.01%) | $0.49_{\pm0.02}\to\mathbf{0.48}_{\pm0.01}$ (4.13%) | $0.43_{\pm0.02}\to\mathbf{0.40}_{\pm0.01}$ (5.05%) | $0.85_{\pm0.02}\to\mathbf{0.79}_{\pm0.01}$ (6.70%) | $0.55_{\pm0.02}\to\mathbf{0.52}_{\pm0.01}$ (1.31%) | $0.43_{\pm0.02}\to\mathbf{0.42}_{\pm0.01}$ (0.44%) |
|  | $0.65_{\pm0.01}\to\mathbf{0.55}_{\pm0.01}$ (15.48%) | $0.60_{\pm0.01}\to\mathbf{0.57}_{\pm0.01}$ (3.7%) | $0.60_{\pm0.01}\to\mathbf{0.57}_{\pm0.01}$ (4.0%) | $0.60_{\pm0.01}\to\mathbf{0.57}_{\pm0.01}$ (4.2%) | $0.60_{\pm0.01}\to\mathbf{0.57}_{\pm0.01}$ (3.8%) | $0.60_{\pm0.01}\to\mathbf{0.57}_{\pm0.01}$ (3.8%) | $0.60_{\pm0.01}\to\mathbf{0.57}_{\pm0.01}$ (3.9%) |
| ETTm1 | $0.43_{\pm0.02}\to\mathbf{0.40}_{\pm0.01}$ (3.73%) | $0.39_{\pm0.02}\to\mathbf{0.38}_{\pm0.01}$ (2.60%) | $0.21_{\pm0.02}\to\mathbf{0.20}_{\pm0.01}$ (2.52%) | $0.36_{\pm0.02}\to\mathbf{0.34}_{\pm0.01}$ (4.53%) | $0.37_{\pm0.02}\to\mathbf{0.35}_{\pm0.01}$ (5.05%) | $0.35_{\pm0.02}\to\mathbf{0.34}_{\pm0.01}$ (2.71%) | $0.71_{\pm0.02}\to\mathbf{0.64}_{\pm0.01}$ (9.98%) |
|  | $0.67_{\pm0.03}\to\mathbf{0.62}_{\pm0.02}$ (6.31%) | $0.53_{\pm0.03}\to\mathbf{0.50}_{\pm0.02}$ (5.24%) | $0.27_{\pm0.03}\to\mathbf{0.25}_{\pm0.02}$ (2.42%) | $0.38_{\pm0.03}\to\mathbf{0.37}_{\pm0.02}$ (2.27%) | $0.40_{\pm0.03}\to\mathbf{0.38}_{\pm0.02}$ (3.62%) | $0.38_{\pm0.03}\to\mathbf{0.37}_{\pm0.02}$ (2.34%) | $0.72_{\pm0.03}\to\mathbf{0.70}_{\pm0.02}$ (2.92%) |
|  | $0.73_{\pm0.02}\to\mathbf{0.68}_{\pm0.01}$ (7.29%) | $0.72_{\pm0.02}\to\mathbf{0.68}_{\pm0.01}$ (5.14%) | $0.33_{\pm0.02}\to\mathbf{0.31}_{\pm0.01}$ (5.96%) | $0.41_{\pm0.02}\to\mathbf{0.40}_{\pm0.01}$ (0.84%) | $0.42_{\pm0.02}\to\mathbf{0.40}_{\pm0.01}$ (3.66%) | $0.50_{\pm0.02}\to\mathbf{0.47}_{\pm0.01}$ (3.7%) | $0.73_{\pm0.02}\to\mathbf{0.71}_{\pm0.01}$ (2.37%) |
|  | $0.78_{\pm0.02}\to\mathbf{0.70}_{\pm0.01}$ (9.83%) | $0.96_{\pm0.02}\to\mathbf{0.85}_{\pm0.01}$ (0.89%) | $0.42_{\pm0.02}\to\mathbf{0.39}_{\pm0.01}$ (0.88%) | $0.46_{\pm0.02}\to\mathbf{0.45}_{\pm0.01}$ (2.8%) | $0.48_{\pm0.02}\to\mathbf{0.46}_{\pm0.01}$ (3.97%) | $0.55_{\pm0.02}\to\mathbf{0.52}_{\pm0.01}$ (2.7%) | $0.84_{\pm0.02}\to\mathbf{0.77}_{\pm0.01}$ (7.56%) |
|  | $0.60_{\pm0.01}\to\mathbf{0.57}_{\pm0.01}$ (7.37%) | $0.60_{\pm0.01}\to\mathbf{0.57}_{\pm0.01}$ (3.7%) | $0.60_{\pm0.01}\to\mathbf{0.57}_{\pm0.01}$ (4.0%) | $0.60_{\pm0.01}\to\mathbf{0.57}_{\pm0.01}$ (4.2%) | $0.60_{\pm0.01}\to\mathbf{0.57}_{\pm0.01}$ (3.8%) | $0.60_{\pm0.01}\to\mathbf{0.57}_{\pm0.01}$ (3.8%) | $0.60_{\pm0.01}\to\mathbf{0.57}_{\pm0.01}$ (3.9%) |
| ETTm2 | $0.27_{\pm0.02}\to\mathbf{0.24}_{\pm0.01}$ (7.87%) | $0.33_{\pm0.02}\to\mathbf{0.31}_{\pm0.01}$ (4.01%) | $0.46_{\pm0.02}\to\mathbf{0.43}_{\pm0.01}$ (3.73%) | $0.20_{\pm0.02}\to\mathbf{0.19}_{\pm0.01}$ (3.99%) | $0.21_{\pm0.02}\to\mathbf{0.19}_{\pm0.01}$ (8.00%) | $0.19_{\pm0.02}\to\mathbf{0.18}_{\pm0.01}$ (3.24%) | $0.25_{\pm0.02}\to\mathbf{0.23}_{\pm0.01}$ (5.35%) |
|  | $0.42_{\pm0.03}\to\mathbf{0.32}_{\pm0.02}$ (22.58%) | $0.87_{\pm0.03}\to\mathbf{0.45}_{\pm0.02}$ (4.8%) | $0.48_{\pm0.03}\to\mathbf{0.45}_{\pm0.02}$ (6.18%) | $0.26_{\pm0.03}\to\mathbf{0.24}_{\pm0.02}$ (7.69%) | $0.31_{\pm0.03}\to\mathbf{0.25}_{\pm0.02}$ (18.97%) | $0.25_{\pm0.03}\to\mathbf{0.24}_{\pm0.02}$ (5.20%) | $0.31_{\pm0.03}\to\mathbf{0.28}_{\pm0.02}$ (7.88%) |
|  | $0.46_{\pm0.02}\to\mathbf{0.35}_{\pm0.01}$ (23.54%) | $0.99_{\pm0.02}\to\mathbf{0.47}_{\pm0.01}$ (3.6%) | $0.50_{\pm0.02}\to\mathbf{0.47}_{\pm0.01}$ (5.96%) | $0.32_{\pm0.02}\to\mathbf{0.29}_{\pm0.01}$ (7.82%) | $0.38_{\pm0.02}\to\mathbf{0.33}_{\pm0.01}$ (13.81%) | $0.50_{\pm0.02}\to\mathbf{0.47}_{\pm0.01}$ (3.7%) | $0.37_{\pm0.02}\to\mathbf{0.34}_{\pm0.01}$ (7.04%) |
|  | $1.15_{\pm0.02}\to\mathbf{0.84}_{\pm0.01}$ (27.00%) | $0.55_{\pm0.02}\to\mathbf{0.52}_{\pm0.01}$ (3.1%) | $0.55_{\pm0.02}\to\mathbf{0.52}_{\pm0.01}$ (7.51%) | $0.41_{\pm0.02}\to\mathbf{0.38}_{\pm0.01}$ (7.36%) | $0.55_{\pm0.02}\to\mathbf{0.52}_{\pm0.01}$ (6.78%) | $0.55_{\pm0.02}\to\mathbf{0.52}_{\pm0.01}$ (2.7%) | $0.47_{\pm0.02}\to\mathbf{0.43}_{\pm0.01}$ (6.99.9%) |
|  | $0.60_{\pm0.01}\to\mathbf{0.57}_{\pm0.01}$ (20.25%) | $3.82_{\pm0.01}\to\mathbf{3.81}_{\pm0.01}$ (3.7%) | $0.60_{\pm0.01}\to\mathbf{0.57}_{\pm0.01}$ (4.0%) | $0.60_{\pm0.01}\to\mathbf{0.57}_{\pm0.01}$ (4.2%) | $0.60_{\pm0.01}\to\mathbf{0.57}_{\pm0.01}$ (3.8%) | $0.60_{\pm0.01}\to\mathbf{0.57}_{\pm0.01}$ (3.8%) | $0.60_{\pm0.01}\to\mathbf{0.57}_{\pm0.01}$ (3.9%) |
| Weather | $0.46_{\pm0.02}\to\mathbf{0.43}_{\pm0.01}$ (6.2%) | $0.46_{\pm0.02}\to\mathbf{0.43}_{\pm0.01}$ (2.5%) | $0.19_{\pm0.02}\to\mathbf{0.18}_{\pm0.01}$ (7.64%) | $0.21_{\pm0.02}\to\mathbf{0.19}_{\pm0.01}$ (5.13%) | $0.46_{\pm0.02}\to\mathbf{0.43}_{\pm0.01}$ (8.88%) | $0.46_{\pm0.02}\to\mathbf{0.43}_{\pm0.01}$ (6.0%) | $0.46_{\pm0.02}\to\mathbf{0.43}_{\pm0.01}$ (5.1%) |
|  | $0.48_{\pm0.03}\to\mathbf{0.45}_{\pm0.02}$ (5.4%) | $0.48_{\pm0.03}\to\mathbf{0.45}_{\pm0.02}$ (4.8%) | $0.23_{\pm0.03}\to\mathbf{0.21}_{\pm0.02}$ (7.89%) | $0.23_{\pm0.03}\to\mathbf{0.21}_{\pm0.02}$ (6.09%) | $0.24_{\pm0.03}\to\mathbf{0.22}_{\pm0.02}$ (8.14%) | $0.48_{\pm0.03}\to\mathbf{0.45}_{\pm0.02}$ (5.0%) | $0.48_{\pm0.03}\to\mathbf{0.45}_{\pm0.02}$ (5.3%) |
|  | $0.50_{\pm0.02}\to\mathbf{0.47}_{\pm0.01}$ (4.1%) | $0.50_{\pm0.02}\to\mathbf{0.47}_{\pm0.01}$ (3.6%) | $0.50_{\pm0.02}\to\mathbf{0.47}_{\pm0.01}$ (6.57%) | $0.29_{\pm0.02}\to\mathbf{0.27}_{\pm0.01}$ (5.55%) | $0.29_{\pm0.02}\to\mathbf{0.27}_{\pm0.01}$ (5.61%) | $0.50_{\pm0.02}\to\mathbf{0.47}_{\pm0.01}$ (3.7%) | $0.50_{\pm0.02}\to\mathbf{0.47}_{\pm0.01}$ (3.8%) |
|  | $0.55_{\pm0.02}\to\mathbf{0.52}_{\pm0.01}$ (2.4%) | $0.55_{\pm0.02}\to\mathbf{0.52}_{\pm0.01}$ (3.1%) | $0.55_{\pm0.02}\to\mathbf{0.52}_{\pm0.01}$ (8.92%) | $0.55_{\pm0.02}\to\mathbf{0.52}_{\pm0.01}$ (7.44%) | $0.35_{\pm0.02}\to\mathbf{0.33}_{\pm0.01}$ (4.07%) | $0.55_{\pm0.02}\to\mathbf{0.52}_{\pm0.01}$ (2.7%) | $0.55_{\pm0.02}\to\mathbf{0.52}_{\pm0.01}$ (2.9%) |
|  | $0.60_{\pm0.01}\to\mathbf{0.57}_{\pm0.01}$ | $0.60_{\pm0.01}\to\mathbf{0.57}_{\pm0.01}$ | $0.60_{\pm0.01}\to\mathbf{0.57}_{\pm0.01}$ | $0.36_{\pm0.01}\to\mathbf{0.33}_{\pm0.01}$ | $0.60_{\pm0.01}\to\mathbf{0.57}_{\pm0.01}$ | $0.60_{\pm0.01}\to\mathbf{0.57}_{\pm0.01}$ | $0.60_{\pm0.01}\to\mathbf{0.57}_{\pm0.01}$ |

| | (4.4%) | (3.7%) | (4.0%) | (4.2%) | (3.8%) | (3.8%) | (3.9%) |
|---|---|---|---|---|---|---|---|
| **Dominick** | $1.22_{\pm0.02} \to 1.10_{\pm0.01}$ (12.93%) | $1.25_{\pm0.02} \to 1.25_{\pm0.01}$ (0.00%) | $1.40_{\pm0.02} \to 1.40_{\pm0.01}$ (0.00%) | $1.37_{\pm0.02} \to 1.37_{\pm0.01}$ (0.00%) | $1.34_{\pm0.02} \to 1.24_{\pm0.01}$ (6.73%) | $1.80_{\pm0.02} \to 1.39_{\pm0.01}$ (22.76%) | $1.26_{\pm1.23} \to 0.43_{\pm0.01}$ (2.35%) |
| | $1.31_{\pm0.03} \to 1.18_{\pm0.02}$ (29.38%) | $0.97_{\pm0.03} \to 0.97_{\pm0.02}$ (0.00%) | $1.05_{\pm0.03} \to 1.05_{\pm0.02}$ (0.00%) | $1.08_{\pm0.03} \to 1.08_{\pm0.02}$ (0.00%) | $1.03_{\pm0.03} \to 0.94_{\pm0.02}$ (8.10%) | $1.66_{\pm0.03} \to 1.20_{\pm0.02}$ (27.52%) | $0.92_{\pm0.03} \to 0.88_{\pm0.02}$ (3.81%) |
| | $1.43_{\pm0.02} \to 1.01_{\pm0.01}$ (9.76%) | $1.09_{\pm0.02} \to 1.06_{\pm0.01}$ (2.56%) | $1.23_{\pm0.02} \to 1.23_{\pm0.01}$ (0.00%) | $1.23_{\pm0.02} \to 1.13_{\pm0.01}$ (8.01%) | $1.25_{\pm0.02} \to 1.13_{\pm0.01}$ (9.08%) | $1.97_{\pm0.02} \to 1.39_{\pm0.01}$ (28.94%) | $1.15_{\pm0.02} \to 1.09_{\pm0.01}$ (3.81%) |
| | $1.55_{\pm0.02} \to 1.34_{\pm0.01}$ (9.61%) | $1.17_{\pm0.02} \to 1.15_{\pm0.01}$ (1.41%) | $1.30_{\pm0.02} \to 1.25_{\pm0.01}$ (3.56%) | $1.32_{\pm0.02} \to 1.22_{\pm0.01}$ (7.34%) | $1.32_{\pm0.02} \to 1.21_{\pm0.01}$ (7.97%) | $2.28_{\pm0.02} \to 1.60_{\pm0.01}$ (29.63%) | $1.26_{\pm0.02} \to 1.18_{\pm0.01}$ (5.68%) |
| | $1.38_{\pm0.01} \to 1.16_{\pm0.01}$ (15.42%) | $1.12_{\pm0.01} \to 1.10_{\pm0.01}$ (1.00%) | $1.25_{\pm0.01} \to 1.23_{\pm0.01}$ (0.89%) | $1.25_{\pm0.01} \to 1.20_{\pm0.01}$ (3.83%) | $1.24_{\pm0.01} \to 1.13_{\pm0.01}$ (7.97%) | $1.92_{\pm0.01} \to 1.40_{\pm0.01}$ (27.21%) | $1.14_{\pm0.01} \to 1.09_{\pm0.01}$ (4.28%) |
| **Human** | $0.37_{\pm0.02} \to 0.31_{\pm0.01}$ (14.19%) | $0.26_{\pm0.02} \to 0.22_{\pm0.01}$ (13.66%) | $0.18_{\pm0.02} \to 0.12_{\pm0.01}$ (32.74%) | $0.17_{\pm0.02} \to 0.11_{\pm0.01}$ (35.22%) | $0.36_{\pm0.02} \to 0.28_{\pm0.01}$ (20.22%) | $0.30_{\pm0.02} \to 0.25_{\pm0.01}$ (15.68%) | $0.13_{\pm0.02} \to 0.14_{\pm0.01}$ (-2.76%) |
| | $0.32_{\pm0.03} \to 0.26_{\pm0.02}$ (16.59%) | $0.26_{\pm0.03} \to 0.22_{\pm0.02}$ (11.86%) | $0.17_{\pm0.03} \to 0.12_{\pm0.02}$ (24.88%) | $0.14_{\pm0.03} \to 0.11_{\pm0.02}$ (17.79%) | $0.52_{\pm0.03} \to 0.24_{\pm0.02}$ (52.35%) | $0.28_{\pm0.03} \to 0.20_{\pm0.02}$ (26.57%) | $0.19_{\pm0.03} \to 0.14_{\pm0.02}$ (25.19%) |
| | $0.30_{\pm0.02} \to 0.17_{\pm0.01}$ (40.52%) | $0.29_{\pm0.02} \to 0.24_{\pm0.01}$ (16.28%) | $0.48_{\pm0.02} \to 0.19_{\pm0.01}$ (58.95%) | $0.52_{\pm0.02} \to 0.16_{\pm0.01}$ (68.20%) | $1.01_{\pm0.02} \to 0.28_{\pm0.01}$ (71.56%) | $0.56_{\pm0.02} \to 0.26_{\pm0.01}$ (52.10%) | $0.26_{\pm0.02} \to 0.19_{\pm0.01}$ (25.10%) |
| | $0.62_{\pm0.02} \to 0.56_{\pm0.01}$ (9.67%) | $0.40_{\pm0.02} \to 0.35_{\pm0.01}$ (11.58%) | $1.60_{\pm0.02} \to 0.18_{\pm0.01}$ (88.50%) | $1.66_{\pm0.02} \to 0.23_{\pm0.01}$ (85.76%) | $1.97_{\pm0.02} \to 0.25_{\pm0.01}$ (87.30%) | $1.27_{\pm0.02} \to 0.28_{\pm0.01}$ (77.75%) | $0.64_{\pm0.02} \to 0.38_{\pm0.01}$ (40.17%) |
| | $0.40_{\pm0.01} \to 0.33_{\pm0.01}$ (20.24%) | $0.30_{\pm0.01} \to 0.25_{\pm0.01}$ (13.34%) | $0.60_{\pm0.01} \to 0.15_{\pm0.01}$ (51.26%) | $0.62_{\pm0.01} \to 0.15_{\pm0.01}$ (51.74%) | $0.96_{\pm0.01} \to 0.26_{\pm0.01}$ (57.85%) | $0.60_{\pm0.01} \to 0.25_{\pm0.01}$ (43.02%) | $0.30_{\pm0.01} \to 0.21_{\pm0.01}$ (21.93%) |
| **KDD** | $1.06_{\pm0.02} \to 0.83_{\pm0.01}$ (20.98%) | $0.71_{\pm0.02} \to 0.70_{\pm0.01}$ (0.19%) | $0.90_{\pm0.02} \to 0.74_{\pm0.01}$ (17.14%) | $0.94_{\pm0.02} \to 0.76_{\pm0.01}$ (19.18%) | $0.84_{\pm0.02} \to 0.77_{\pm0.01}$ (7.81%) | $0.92_{\pm0.02} \to 0.76_{\pm0.01}$ (16.63%) | $0.96_{\pm0.02} \to 0.77_{\pm0.01}$ (19.59%) |
| | $1.20_{\pm0.03} \to 0.94_{\pm0.02}$ (21.11%) | $0.81_{\pm0.03} \to 0.81_{\pm0.02}$ (-0.06%) | $1.08_{\pm0.03} \to 0.86_{\pm0.02}$ (19.98%) | $1.11_{\pm0.03} \to 0.88_{\pm0.02}$ (20.64%) | $0.93_{\pm0.03} \to 0.86_{\pm0.02}$ (7.24%) | $1.08_{\pm0.03} \to 0.87_{\pm0.02}$ (18.61%) | $1.08_{\pm0.03} \to 0.85_{\pm0.02}$ (20.86%) |
| | $1.29_{\pm0.02} \to 0.98_{\pm0.01}$ (23.58%) | $0.84_{\pm0.02} \to 0.84_{\pm0.01}$ (-0.16%) | $1.19_{\pm0.02} \to 0.93_{\pm0.01}$ (21.71%) | $1.21_{\pm0.02} \to 0.93_{\pm0.01}$ (22.58%) | $0.95_{\pm0.02} \to 0.87_{\pm0.01}$ (7.89%) | $1.17_{\pm0.02} \to 0.93_{\pm0.01}$ (19.71%) | $1.18_{\pm0.02} \to 0.96_{\pm0.01}$ (23.52%) |
| | $1.47_{\pm0.02} \to 1.05_{\pm0.01}$ (28.32%) | $0.83_{\pm0.02} \to 0.83_{\pm0.01}$ (0.00%) | $1.33_{\pm0.02} \to 0.99_{\pm0.01}$ (25.51%) | $1.31_{\pm0.02} \to 0.88_{\pm0.01}$ (24.92%) | $1.01_{\pm0.02} \to 0.89_{\pm0.01}$ (11.22%) | $1.24_{\pm0.02} \to 0.97_{\pm0.01}$ (21.63%) | $1.28_{\pm0.02} \to 0.96_{\pm0.01}$ (24.66%) |
| | $1.25_{\pm0.01} \to 0.95_{\pm0.01}$ (23.50%) | $0.80_{\pm0.01} \to 0.80_{\pm0.01}$ (-0.03%) | $1.13_{\pm0.01} \to 0.88_{\pm0.01}$ (21.09%) | $1.14_{\pm0.01} \to 0.89_{\pm0.01}$ (21.83%) | $0.93_{\pm0.01} \to 0.85_{\pm0.01}$ (8.54%) | $1.10_{\pm0.01} \to 0.88_{\pm0.01}$ (19.14%) | $1.12_{\pm0.01} \to 0.87_{\pm0.01}$ (22.15%) |
| **Nature** | $1.22_{\pm0.02} \to 0.79_{\pm0.01}$ (35.00%) | $0.28_{\pm0.02} \to 0.28_{\pm0.01}$ (-1.74%) | $0.29_{\pm0.02} \to 0.28_{\pm0.01}$ (1.95%) | $0.27_{\pm0.02} \to 0.26_{\pm0.01}$ (1.37%) | $0.87_{\pm0.02} \to 0.80_{\pm0.01}$ (7.16%) | $1.04_{\pm0.02} \to 0.83_{\pm0.01}$ (19.90%) | $0.78_{\pm0.02} \to 0.7_{\pm0.01}$ (0.38%) |
| | $1.74_{\pm0.03} \to 1.19_{\pm0.02}$ (31.15%) | $0.63_{\pm0.03} \to 0.66_{\pm0.02}$ (-3.91%) | $0.30_{\pm0.03} \to 0.29_{\pm0.02}$ (2.83%) | $0.24_{\pm0.03} \to 0.23_{\pm0.02}$ (2.38%) | $0.99_{\pm0.03} \to 0.95_{\pm0.02}$ (3.25%) | $1.05_{\pm0.03} \to 0.97_{\pm0.02}$ (6.86%) | $0.96_{\pm0.03} \to 0.95_{\pm0.02}$ (0.39%) |
| | $1.01_{\pm0.02} \to 0.95_{\pm0.01}$ (5.80%) | $0.83_{\pm0.02} \to 0.81_{\pm0.01}$ (1.10%) | $0.30_{\pm0.02} \to 0.28_{\pm0.01}$ (3.99%) | $0.22_{\pm0.02} \to 0.20_{\pm0.01}$ (4.75%) | $0.98_{\pm0.02} \to 0.94_{\pm0.01}$ (3.19%) | $1.04_{\pm0.02} \to 0.98_{\pm0.01}$ (5.33%) | $0.95_{\pm0.02} \to 0.94_{\pm0.01}$ (0.62%) |
| | $1.01_{\pm0.02} \to 0.93_{\pm0.01}$ (7.44%) | $0.88_{\pm0.02} \to 0.87_{\pm0.01}$ (-0.13%) | $0.49_{\pm0.02} \to 0.47_{\pm0.01}$ (4.30%) | $0.34_{\pm0.02} \to 0.31_{\pm0.01}$ (6.40%) | $0.99_{\pm0.02} \to 0.95_{\pm0.01}$ (3.65%) | $0.98_{\pm0.02} \to 0.95_{\pm0.01}$ (2.61%) | $0.96_{\pm0.02} \to 0.94_{\pm0.01}$ (2.06%) |
| | $1.25_{\pm0.01} \to 0.96_{\pm0.01}$ (21.69%) | $0.65_{\pm0.01} \to 0.66_{\pm0.01}$ (-1.17%) | $0.34_{\pm0.01} \to 0.33_{\pm0.01}$ (3.26%) | $0.26_{\pm0.01} \to 0.25_{\pm0.01}$ (3.72%) | $0.95_{\pm0.01} \to 0.91_{\pm0.01}$ (4.31%) | $1.02_{\pm0.01} \to 0.94_{\pm0.01}$ (8.67%) | $0.91_{\pm0.01} \to 0.90_{\pm0.01}$ (0.86%) |
| **NASDAQ** | $0.72_{\pm0.02} \to 0.55_{\pm0.01}$ (23.38%) | $0.36_{\pm0.02} \to 0.36_{\pm0.01}$ (-1.05%) | $0.51_{\pm0.02} \to 0.42_{\pm0.01}$ (16.14%) | $0.50_{\pm0.02} \to 0.42_{\pm0.01}$ (14.96%) | $0.56_{\pm0.02} \to 0.53_{\pm0.01}$ (5.24%) | $0.51_{\pm0.02} \to 0.43_{\pm0.01}$ (15.21%) | $0.57_{\pm0.02} \to 0.48_{\pm0.01}$ (15.20%) |
| | $0.78_{\pm0.03} \to 0.62_{\pm0.02}$ (19.76%) | $0.41_{\pm0.03} \to 0.41_{\pm0.02}$ (-0.24%) | $0.70_{\pm0.03} \to 0.56_{\pm0.02}$ (18.99%) | $0.67_{\pm0.03} \to 0.54_{\pm0.02}$ (18.08%) | $0.67_{\pm0.03} \to 0.62_{\pm0.02}$ (6.62%) | $0.67_{\pm0.03} \to 0.54_{\pm0.02}$ (18.34%) | $0.75_{\pm0.03} \to 0.61_{\pm0.02}$ (17.99%) |
| | $0.98_{\pm0.02} \to 0.76_{\pm0.01}$ (21.70%) | $0.51_{\pm0.02} \to 0.51_{\pm0.01}$ (-0.09%) | $0.88_{\pm0.02} \to 0.70_{\pm0.01}$ (19.70%) | $0.87_{\pm0.02} \to 0.70_{\pm0.01}$ (19.63%) | $0.82_{\pm0.02} \to 0.75_{\pm0.01}$ (8.13%) | $0.87_{\pm0.02} \to 0.70_{\pm0.01}$ (19.47%) | $0.96_{\pm0.02} \to 0.91_{\pm0.01}$ (21.48%) |
| | $1.18_{\pm0.02} \to 0.88_{\pm0.01}$ (24.72%) | $0.57_{\pm0.02} \to 0.56_{\pm0.01}$ (0.06%) | $0.80_{\pm0.02} \to 0.63_{\pm0.01}$ (24.26%) | $1.12_{\pm0.02} \to 0.83_{\pm0.01}$ (24.58%) | $0.93_{\pm0.02} \to 0.83_{\pm0.01}$ (10.40%) | $1.10_{\pm0.02} \to 0.83_{\pm0.01}$ (23.74%) | $1.14_{\pm0.02} \to 1.11_{\pm0.01}$ (23.76%) |
| | $0.91_{\pm0.01} \to 0.70_{\pm0.01}$ (22.39%) | $0.46_{\pm0.01} \to 0.46_{\pm0.01}$ (-0.33%) | $0.80_{\pm0.01} \to 0.63_{\pm0.01}$ (19.77%) | $0.79_{\pm0.01} \to 0.68_{\pm0.01}$ (14.76%) | $0.74_{\pm0.01} \to 0.68_{\pm0.01}$ (7.59%) | $0.78_{\pm0.01} \to 0.62_{\pm0.01}$ (19.19%) | $0.85_{\pm0.01} \to 0.79_{\pm0.01}$ (8.96%) |
| **Pedestrian** | $0.38_{\pm0.02} \to 0.22_{\pm0.01}$ (41.72%) | $0.11_{\pm0.02} \to 0.11_{\pm0.01}$ (-0.03%) | $0.09_{\pm0.02} \to 0.08_{\pm0.01}$ (2.85%) | $0.14_{\pm0.02} \to 0.13_{\pm0.01}$ (3.68%) | $0.33_{\pm0.02} \to 0.27_{\pm0.01}$ (18.21%) | $0.14_{\pm0.02} \to 0.13_{\pm0.01}$ (1.53%) | $0.21_{\pm0.02} \to 0.18_{\pm0.01}$ (18.38%) |
| | $0.38_{\pm0.03} \to 0.24_{\pm0.02}$ (34.76%) | $0.13_{\pm0.03} \to 0.12_{\pm0.02}$ (0.43%) | $0.14_{\pm0.03} \to 0.12_{\pm0.02}$ (11.79%) | $0.22_{\pm0.03} \to 0.18_{\pm0.02}$ (11.82%) | $0.74_{\pm0.03} \to 0.63_{\pm0.02}$ (14.79%) | $0.20_{\pm0.03} \to 0.18_{\pm0.02}$ (9.37%) | $0.39_{\pm0.03} \to 0.33_{\pm0.02}$ (15.51%) |
| | $0.52_{\pm0.02} \to 0.30_{\pm0.01}$ | $0.15_{\pm0.02} \to 0.14_{\pm0.01}$ | $0.16_{\pm0.02} \to 0.14_{\pm0.01}$ | $0.25_{\pm0.02} \to 0.22_{\pm0.01}$ | $0.77_{\pm0.02} \to 0.66_{\pm0.01}$ | $0.22_{\pm0.02} \to 0.20_{\pm0.01}$ | $0.35_{\pm0.02} \to 0.29_{\pm0.01}$ |

| | | | | | | | |
|---|---|---|---|---|---|---|---|
| | (41.30%) | (0.37%) | (12.15%) | (11.49%) | (13.38%) | (8.83%) | (18.22%) |
| | $0.55_{\pm0.02} \to \mathbf{0.31}_{\pm0.01}$ (42.27%) | $0.16_{\pm0.02} \to \mathbf{0.15}_{\pm0.01}$ (0.30%) | $0.18_{\pm0.02} \to \mathbf{0.16}_{\pm0.01}$ (10.94%) | $0.27_{\pm0.02} \to \mathbf{0.25}_{\pm0.01}$ (11.18%) | $0.77_{\pm0.02} \to \mathbf{0.66}_{\pm0.01}$ (13.88%) | $0.25_{\pm0.02} \to \mathbf{0.22}_{\pm0.01}$ (8.01%) | $0.38_{\pm0.02} \to \mathbf{0.31}_{\pm0.01}$ (19.35%) |
| | $0.46_{\pm0.01} \to \mathbf{0.27}_{\pm0.01}$ (40.01%) | $0.14_{\pm0.01} \to \mathbf{0.13}_{\pm0.01}$ (0.26%) | $0.14_{\pm0.01} \to \mathbf{0.12}_{\pm0.01}$ (9.43%) | $0.22_{\pm0.01} \to \mathbf{0.19}_{\pm0.01}$ (9.54%) | $0.69_{\pm0.01} \to \mathbf{0.60}_{\pm0.01}$ (14.20%) | $0.20_{\pm0.01} \to \mathbf{0.18}_{\pm0.01}$ (6.94%) | $0.33_{\pm0.01} \to \mathbf{0.27}_{\pm0.01}$ (15.87%) |
| *Tourism* | $0.38_{\pm0.02} \to \mathbf{0.33}_{\pm0.01}$ (12.79%) | $0.11_{\pm0.02} \to \mathbf{0.08}_{\pm0.01}$ (25.00%) | $0.16_{\pm0.02} \to \mathbf{0.11}_{\pm0.01}$ (26.16%) | $0.16_{\pm0.02} \to \mathbf{0.11}_{\pm0.01}$ (28.55%) | $0.34_{\pm0.02} \to \mathbf{0.22}_{\pm0.01}$ (33.74%) | $0.17_{\pm0.02} \to \mathbf{0.08}_{\pm0.01}$ (50.06%) | $0.17_{\pm0.02} \to \mathbf{0.12}_{\pm0.01}$ (27.43%) |
| | $0.13_{\pm0.03} \to \mathbf{0.10}_{\pm0.02}$ (17.05%) | $0.08_{\pm0.03} \to \mathbf{0.07}_{\pm0.02}$ (8.79%) | $0.11_{\pm0.03} \to \mathbf{0.10}_{\pm0.02}$ (8.96%) | $0.12_{\pm0.03} \to \mathbf{0.10}_{\pm0.02}$ (3.10%) | $0.40_{\pm0.03} \to \mathbf{0.17}_{\pm0.02}$ (56.60%) | $0.20_{\pm0.03} \to \mathbf{0.11}_{\pm0.02}$ (43.12%) | $0.14_{\pm0.03} \to \mathbf{0.10}_{\pm0.02}$ (22.82%) |
| | $0.30_{\pm0.02} \to \mathbf{0.31}_{\pm0.01}$ (-3.60%) | $0.15_{\pm0.02} \to \mathbf{0.14}_{\pm0.01}$ (6.10%) | $0.50_{\pm0.02} \to \mathbf{0.26}_{\pm0.01}$ (47.99%) | $0.50_{\pm0.02} \to \mathbf{0.26}_{\pm0.01}$ (47.75%) | $0.68_{\pm0.02} \to \mathbf{0.37}_{\pm0.01}$ (44.29%) | $0.33_{\pm0.02} \to \mathbf{0.15}_{\pm0.01}$ (53.09%) | $0.33_{\pm0.02} \to \mathbf{0.31}_{\pm0.01}$ (4.90%) |
| | $0.18_{\pm0.02} \to \mathbf{0.14}_{\pm0.01}$ (17.30%) | $0.31_{\pm0.02} \to \mathbf{0.29}_{\pm0.01}$ (6.04%) | $0.48_{\pm0.02} \to \mathbf{0.11}_{\pm0.01}$ (77.23%) | $0.49_{\pm0.02} \to \mathbf{0.11}_{\pm0.01}$ (77.41%) | $0.66_{\pm0.02} \to \mathbf{0.25}_{\pm0.01}$ (61.01%) | $0.28_{\pm0.02} \to \mathbf{0.15}_{\pm0.01}$ (46.32%) | $0.26_{\pm0.02} \to \mathbf{0.19}_{\pm0.01}$ (27.16%) |
| | $0.24_{\pm0.01} \to \mathbf{0.22}_{\pm0.01}$ (10.88%) | $0.16_{\pm0.01} \to \mathbf{0.14}_{\pm0.01}$ (11.48%) | $0.31_{\pm0.01} \to \mathbf{0.14}_{\pm0.01}$ (40.08%) | $0.31_{\pm0.01} \to \mathbf{0.14}_{\pm0.01}$ (39.20%) | $0.52_{\pm0.01} \to \mathbf{0.25}_{\pm0.01}$ (48.91%) | $0.24_{\pm0.01} \to \mathbf{0.12}_{\pm0.01}$ (48.14%) | $0.22_{\pm0.01} \to \mathbf{0.18}_{\pm0.01}$ (20.57%) |
| *Vehicle trips* | $1.40_{\pm0.02} \to \mathbf{1.13}_{\pm0.01}$ (19.00%) | $0.64_{\pm0.02} \to \mathbf{0.62}_{\pm0.01}$ (2.71%) | $0.89_{\pm0.02} \to \mathbf{0.78}_{\pm0.01}$ (11.63%) | $0.86_{\pm0.02} \to \mathbf{0.71}_{\pm0.01}$ (16.60%) | $1.24_{\pm0.02} \to \mathbf{1.14}_{\pm0.01}$ (7.46%) | $1.45_{\pm0.02} \to \mathbf{0.72}_{\pm0.01}$ (26.95%) | $1.15_{\pm0.02} \to \mathbf{0.83}_{\pm0.01}$ (8.02%) |
| | $1.39_{\pm0.03} \to \mathbf{1.14}_{\pm0.02}$ (17.78%) | $0.84_{\pm0.03} \to \mathbf{0.85}_{\pm0.02}$ (-2.23%) | $0.81_{\pm0.03} \to \mathbf{0.68}_{\pm0.02}$ (15.09%) | $0.79_{\pm0.03} \to \mathbf{0.69}_{\pm0.02}$ (11.95%) | $1.25_{\pm0.03} \to \mathbf{1.10}_{\pm0.02}$ (11.83%) | $1.37_{\pm0.03} \to \mathbf{0.77}_{\pm0.02}$ (21.34%) | $1.05_{\pm0.03} \to \mathbf{0.81}_{\pm0.02}$ (6.77%) |
| | $1.48_{\pm0.02} \to \mathbf{1.18}_{\pm0.01}$ (19.65%) | $0.95_{\pm0.02} \to \mathbf{0.94}_{\pm0.01}$ (0.82%) | $1.08_{\pm0.02} \to \mathbf{0.87}_{\pm0.01}$ (18.82%) | $1.02_{\pm0.02} \to \mathbf{0.80}_{\pm0.01}$ (20.85%) | $1.34_{\pm0.02} \to \mathbf{1.11}_{\pm0.01}$ (16.47%) | $1.78_{\pm0.02} \to \mathbf{1.28}_{\pm0.01}$ (27.59%) | $1.25_{\pm0.02} \to \mathbf{1.04}_{\pm0.01}$ (16.29%) |
| | $1.30_{\pm0.02} \to \mathbf{1.09}_{\pm0.01}$ (16.09%) | $0.90_{\pm0.02} \to \mathbf{0.87}_{\pm0.01}$ (2.27%) | $1.32_{\pm0.02} \to \mathbf{1.01}_{\pm0.01}$ (23.03%) | $1.27_{\pm0.02} \to \mathbf{0.98}_{\pm0.01}$ (22.63%) | $1.59_{\pm0.02} \to \mathbf{1.24}_{\pm0.01}$ (21.70%) | $2.13_{\pm0.02} \to \mathbf{1.42}_{\pm0.01}$ (33.07%) | $1.40_{\pm0.02} \to \mathbf{1.12}_{\pm0.01}$ (19.63%) |
| | $1.39_{\pm0.01} \to \mathbf{1.13}_{\pm0.01}$ (18.13%) | $0.83_{\pm0.01} \to \mathbf{0.82}_{\pm0.01}$ (0.89%) | $1.02_{\pm0.01} \to \mathbf{0.84}_{\pm0.01}$ (17.14%) | $0.98_{\pm0.01} \to \mathbf{0.80}_{\pm0.01}$ (18.00%) | $1.35_{\pm0.01} \to \mathbf{1.15}_{\pm0.01}$ (14.37%) | $1.68_{\pm0.01} \to \mathbf{1.05}_{\pm0.01}$ (38.46%) | $1.21_{\pm0.01} \to \mathbf{0.95}_{\pm0.01}$ (21.54%) |
| *Weather* | $1.01_{\pm0.02} \to \mathbf{0.88}_{\pm0.01}$ (12.10%) | $0.83_{\pm0.02} \to \mathbf{0.82}_{\pm0.01}$ (0.04%) | $0.96_{\pm0.02} \to \mathbf{0.86}_{\pm0.01}$ (9.69%) | $0.96_{\pm0.02} \to \mathbf{0.87}_{\pm0.01}$ (9.32%) | $0.91_{\pm0.02} \to \mathbf{0.89}_{\pm0.01}$ (2.07%) | $0.93_{\pm0.02} \to \mathbf{0.85}_{\pm0.01}$ (8.54%) | $0.97_{\pm0.02} \to \mathbf{0.87}_{\pm0.01}$ (10.07%) |
| | $1.03_{\pm0.03} \to \mathbf{0.90}_{\pm0.02}$ (12.15%) | $0.86_{\pm0.03} \to \mathbf{0.85}_{\pm0.02}$ (0.03%) | $1.01_{\pm0.03} \to \mathbf{0.89}_{\pm0.02}$ (10.92%) | $1.01_{\pm0.03} \to \mathbf{0.90}_{\pm0.02}$ (10.80%) | $0.93_{\pm0.03} \to \mathbf{0.90}_{\pm0.02}$ (2.18%) | $0.98_{\pm0.03} \to \mathbf{0.88}_{\pm0.02}$ (9.95%) | $1.01_{\pm0.03} \to \mathbf{0.89}_{\pm0.02}$ (11.32%) |
| | $1.13_{\pm0.02} \to \mathbf{0.96}_{\pm0.01}$ (14.46%) | $0.87_{\pm0.02} \to \mathbf{0.86}_{\pm0.01}$ (0.002%) | $1.10_{\pm0.02} \to \mathbf{0.95}_{\pm0.01}$ (13.54%) | $1.10_{\pm0.02} \to \mathbf{0.95}_{\pm0.01}$ (13.48%) | $0.98_{\pm0.02} \to \mathbf{0.94}_{\pm0.01}$ (3.76%) | $1.06_{\pm0.02} \to \mathbf{0.92}_{\pm0.01}$ (12.43%) | $1.12_{\pm0.02} \to \mathbf{0.96}_{\pm0.01}$ (13.90%) |
| | $1.33_{\pm0.02} \to \mathbf{1.08}_{\pm0.01}$ (18.43%) | $0.85_{\pm0.02} \to \mathbf{0.84}_{\pm0.01}$ (0.08%) | $1.34_{\pm0.02} \to \mathbf{1.09}_{\pm0.01}$ (18.15%) | $1.34_{\pm0.02} \to \mathbf{1.09}_{\pm0.01}$ (18.24%) | $1.07_{\pm0.02} \to \mathbf{1.00}_{\pm0.01}$ (5.87%) | $1.29_{\pm0.02} \to \mathbf{1.06}_{\pm0.01}$ (17.63%) | $1.35_{\pm0.02} \to \mathbf{1.09}_{\pm0.01}$ (18.57%) |
| | $1.12_{\pm0.01} \to \mathbf{0.96}_{\pm0.01}$ (14.29%) | $0.85_{\pm0.01} \to \mathbf{0.84}_{\pm0.01}$ (0.03%) | $1.10_{\pm0.01} \to \mathbf{0.95}_{\pm0.01}$ (13.07%) | $1.10_{\pm0.01} \to \mathbf{0.95}_{\pm0.01}$ (12.96%) | $0.97_{\pm0.01} \to \mathbf{0.93}_{\pm0.01}$ (3.47%) | $1.06_{\pm0.01} \to \mathbf{0.93}_{\pm0.01}$ (12.13%) | $1.11_{\pm0.01} \to \mathbf{0.95}_{\pm0.01}$ (13.47%) |

## A.4 EXPERIMENTS ON PERFORMANCE IMPROVEMENT AS A FUNCTION OF THE NUMBER OF ACTIONS

In this section, we analyze the performance improvements achieved by post-training optimization on the *DLinear* model for the `ETTm1` dataset. Specifically, we examine how the performance varies with different prediction horizons and different numbers of actions. The results are summarized in the table below:

Table 8: Adaptive optimization improved performance.

| Horizon | 2 Actions | 4 Actions | 7 Actions |
|---|---|---|---|
| 96 | 3.63 ± 0.54 | 3.74 ± 0.93 | 5.05 ± 0.39 |
| 192 | 2.10 ± 0.26 | 2.80 ± 0.47 | 3.62 ± 0.66 |
| 336 | 3.25 ± 0.93 | 3.45 ± 0.61 | 3.66 ± 0.71 |
| 720 | 2.37 ± 0.78 | 3.29 ± 0.47 | 3.97 ± 0.78 |

The table displays the performance improvements (in terms of error reduction) achieved through adaptive optimization at various prediction horizons (96, 192, 336, and 720) and with different numbers of actions (2, 4, and 7). The values are presented as mean ± standard deviation, providing an indication of the variability in performance.

## A.5 Cross-Metric Evaluation of Optimization Strategies

To systematically investigate whether optimizing a single metric (e.g., Mean Squared Error, MSE) can lead to artificial or misaligned improvements in other metrics, we designed a comprehensive evaluation framework.

### Methodology

We constructed a **cross-metric evaluation matrix** with the following structure:

- **Rows:** Metrics used for optimization, including MSE, Mean Absolute Error (MAE), Mean Absolute Percentage Error (MAPE), and $R^2$.
- **Columns:** Metrics used for evaluation.

For each cell, we report the **relative improvement per episode** across the training, validation, and test sets, quantifying how optimizing one metric affects the others.

### Results and Visualization

Figure 12 provides a visual summary. We observe two interesting phenomena:

- **Consistent landscapes:** Across most optimization settings, the training, validation, and test losses share similar landscapes, indicating that overfitting is empirically mitigated. This aligns with the theoretical rationale that our small, interpretable action set limits overfitting.
- **Cross-metric agreement:** Improvements generalize well across metrics, confirming that our approach is not merely metric-specific gaming but delivers genuine gains across multiple evaluation criteria.

- **Aligned metrics:** When metrics are well-aligned, improvements generally transfer across metrics.
- **Incompatible metrics:** Optimization may slightly degrade performance on incompatible metrics; for example, optimizing MAPE can reduce $R^2$.

## A.6 More experiments on the human feedback

**Detailed principle of the human in the loop framework**    The system generates executable code based on user feedback using a language model (LLM). The process is as follows:

- **User Feedback**: The user provides a natural language description of the desired transformation (e.g., scaling predictions).
- **Prompt Generation**: The feedback is passed through a function that creates a structured prompt for the LLM.
- **LLM Code Generation**: The LLM generates a Python class and function based on the feedback. The class includes a transformation function and a parameter generation function.
- **Optimization**: Following code generation, the system optimizes the transformation via bandit, RL or genetic algorithms to improve performance.

The generated prompt is structured as follows:

```python
def build_feedback_prompt(feedback):
    return f"""
    Given the following feedback about a time series prediction model:

    Feedback: "{feedback}"

    Please generate a Python class called 'GenericFunction' that represents a transfo
```

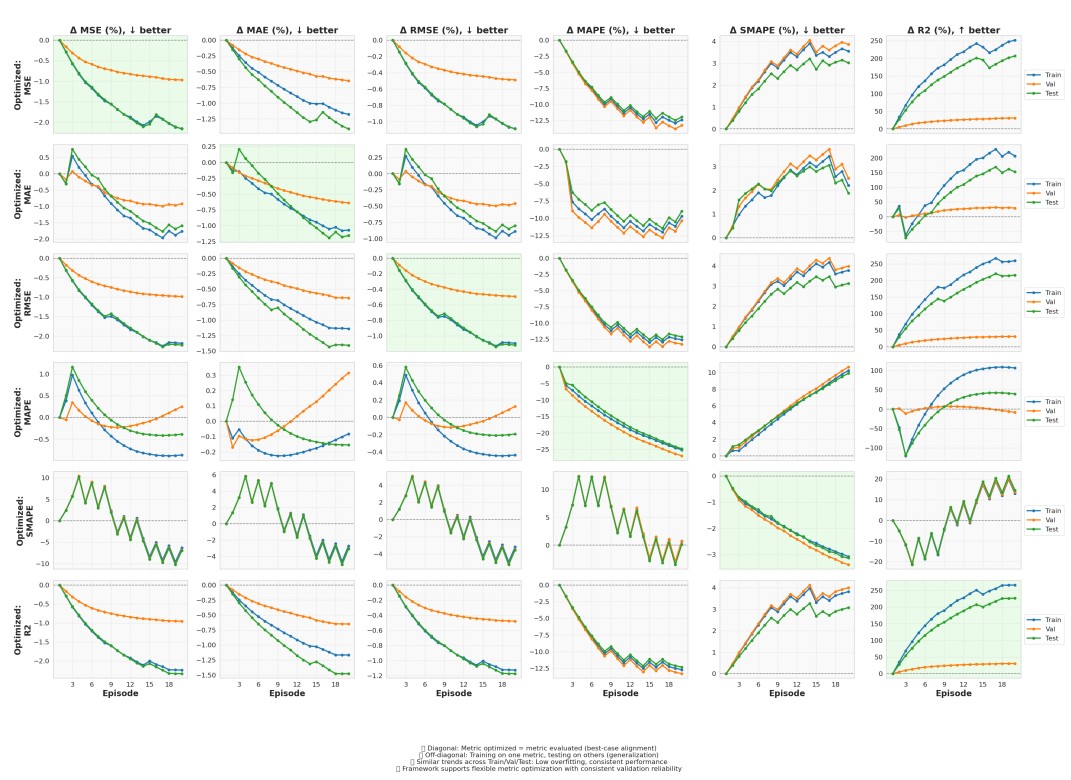

Figure 11: Cross-metric relative improvements. Rows correspond to the optimization metric, columns correspond to the evaluation metric.

```
1. A constructor ('__init__') that accepts:
    - 'function_type': The type of transformation.
    - 'params': A dictionary containing parameters for the transformation.

2. An 'apply' method that modifies the prediction or context vector and outputs

Additionally, generate a function 'generate_random_params_for_action' that retur

--- START OF GENERATED CODE ---

# Class Definition:
class GenericFunction:
<class-body>

# Function Definition:
def generate_random_params_for_action(action, batch_x):
<function-body>

--- END OF GENERATED CODE ---
"""
```

The system then optimizes the generated code using the proposed optimization schemes, ensuring that the generated transformations lead to performance improvements.

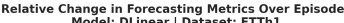

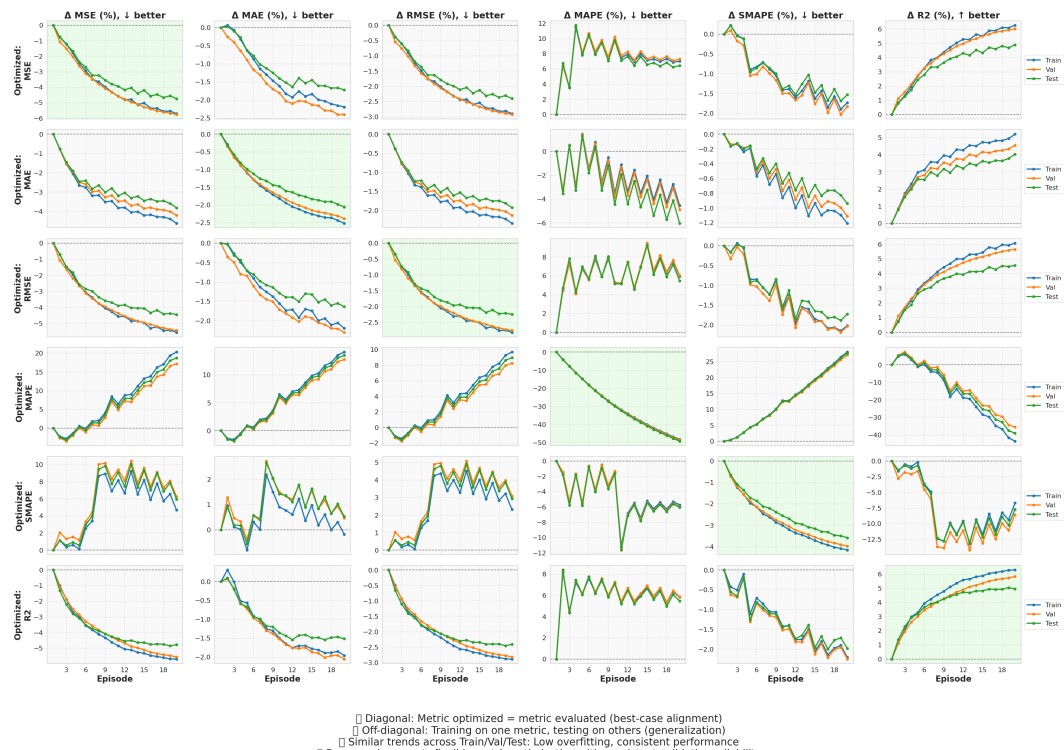

Figure 12: Cross-metric relative improvements. Rows correspond to the optimization metric, columns correspond to the evaluation metric.

### A.6.1 HUMAN FEEDBACK IN ACTION

In this section, we illustrate the practical impact of human feedback through three representative examples, each consisting of a triplet of subplots. These examples demonstrate how natural language insights from a human user can be translated into targeted post-training actions, improving forecasting accuracy beyond automated optimization alone.

Each example includes the following three visualizations:

- **Forecast Comparison with Feedback Summary:** The first subplot presents the full forecasting context: the historical context vector, the model's initial predictions, the predictions after reinforcement learning (RL)-based optimization, and the final forecast incorporating human feedback. The title of each subplot includes the specific textual instruction provided by the human. This view emphasizes how the feedback alters the forecasted trajectory.

- **Generated Code from Human Instruction:** The second subplot displays the code snippet generated by a lightweight language model (LLM) based on the human's textual feedback. This demonstrates the interpretability and direct translatability of natural language instructions into executable post-processing transformations.

- **RMSE Improvement Visualization:** The third subplot shows the reduction in RMSE achieved by applying the human-guided correction compared to the RL-only optimization. This quantifies the value added by the human-in-the-loop mechanism.

Each of the three examples showcases a different type of human insight—such as noise reduction, trend adjustment, or outlier suppression—emphasizing both the flexibility and effectiveness of

incorporating human feedback in the post-training phase. These case studies highlight the potential of combining automated learning with domain expertise to refine time series forecasts in practice.

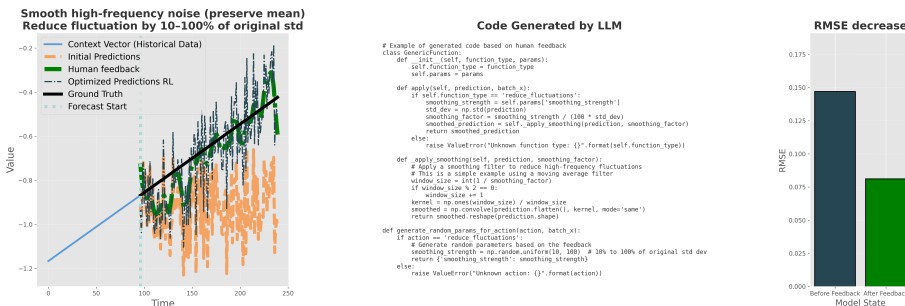

Figure 13: Human feedback integration example 1: Forecast comparison with RL and human feedback (top), code generated from human feedback (middle), and RMSE improvement (bottom). **Dataset:** `Dominick`, **Model:** `PatchTST`

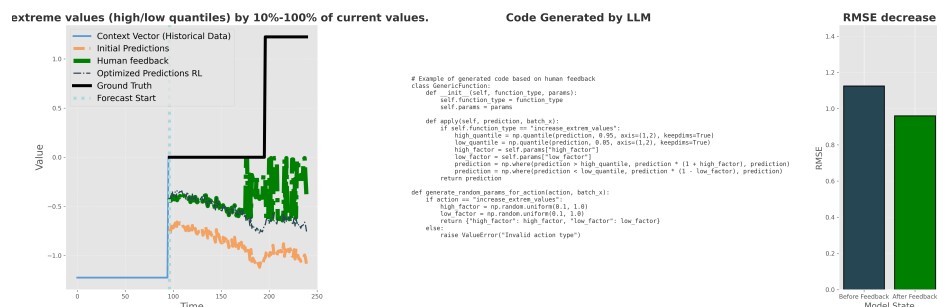

Figure 14: Human feedback integration example 2: Forecast comparison with RL and human feedback (top), code generated from human feedback (middle), and RMSE improvement (bottom). **Dataset:** `Nature`, **Model:** `DLinear`

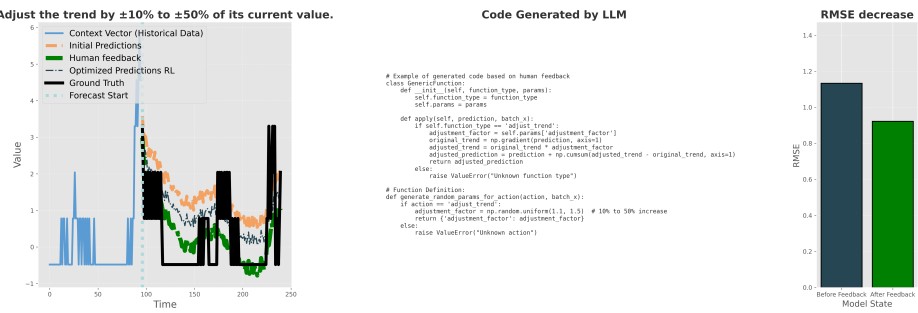

Figure 15: Human feedback integration example 3: Forecast comparison with RL and human feedback (top), code generated from human feedback (middle), and RMSE improvement (bottom). **Dataset:** `Tourism`, **Model:** `PatchTST`

### A.6.2 ROBUSTNESS ANALYSIS WITH RESPECT TO THE PROMPT

To demonstrate the robustness of the proposed framework, we analyze failure cases where the prompt is either ambiguous or nonsensical. These cases are intentionally designed to test the system's ability to handle invalid or poorly defined feedback. The framework is robust in that it identifies and discards actions that do not lead to performance improvements, ensuring that only meaningful transformations are applied.

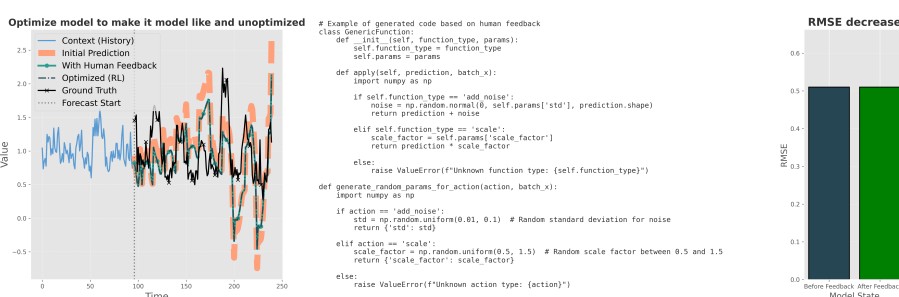

Figure 16: Human feedback integration example 4: Forecast comparison with RL and human feedback (top), code generated from human feedback (middle), and RMSE improvement (bottom). **Dataset:** ETTm1, **Model:** DLinear

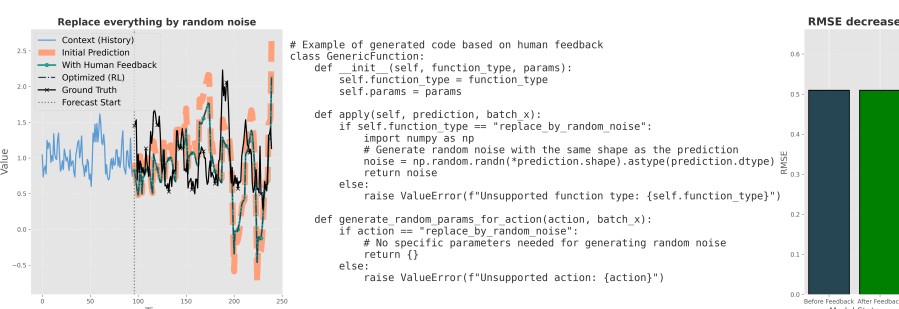

Figure 17: Human feedback integration example 3: Forecast comparison with RL and human feedback (top), code generated from human feedback (middle), and RMSE improvement (bottom). **Dataset:** ETTm1, **Model:** DLinear

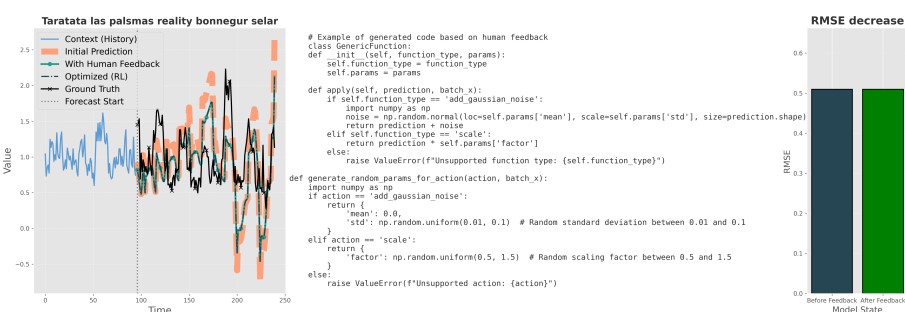

Figure 18: Human feedback integration example 6: Forecast comparison with RL and human feedback (top), code generated from human feedback (middle), and RMSE improvement (bottom). **Dataset:** ETTm1, **Model:** DLinear

## A.7 CODE AND REPRODUCIBILITY

To enable full reproducibility of our results, we provide detailed instructions for using the code associated with our framework. This section includes guidelines for setting up the environment, running the experiments, and utilizing the graphical user interface (GUI) for easy interaction with the framework. We also provide links to the repository, ensuring that interested readers can freely access and experiment with our code.

## A.8 CODE USAGE AND API DOCUMENTATION

This appendix provides instructions for using the codebase and the API for time series model post-training and human feedback exploration. The framework provides a method for users to adjust model predictions using human feedback and contextual bandit algorithms, allowing the model to dynamically adapt its behavior. The code is available at `https://github.com/posttraining/post_training`.

### GOAL

The primary goal of this project is to provide an interactive environment where users can fine-tune time series model predictions based on human feedback. The framework leverages a contextual bandit approach, allowing users to explore different actions and see their impact on the model's predictions.

### FEATURES

- **Time Series Model Exploration:** Train and explore various time series models with different parameters and datasets.

- **Optimization Framework:** Dynamically apply actions and evaluate their effects on the model's prediction accuracy.

- **Human Feedback Integration:** Users can provide feedback on the predictions to improve the model's output over time.

- **Streamlit Interface:** An interactive frontend for exploring and providing feedback on model predictions.

### A.8.1 EXAMPLES TO USE THE STREAMLIT APPLICATION

To experiment with the Streamlit application, follow these steps:

1. **Click on the following (link):** Go to the webpage. You should see the configuration page as in Figure 19.

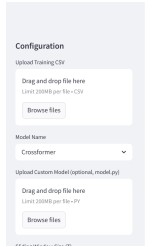
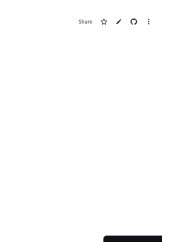

Figure 19: Configuration page

2. **Upload CSV File:** Upload a CSV file containing the time series data. The file should be in CSV format, with rows representing different time steps and columns representing different features for multivariate datasets. A sample file, `train.csv`, is provided in the supplementary materials. You can see an example in Figure 20

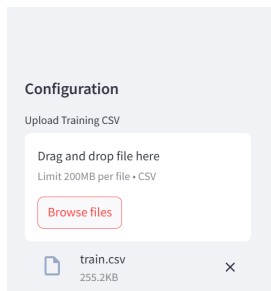

Figure 20: Example of upload file

3. **Select Model and Options**: Choose the model and other options. For the model, use DLinear, as other models require a GPU to run or will take longer. The server currently supports CPU only as in Figure 21

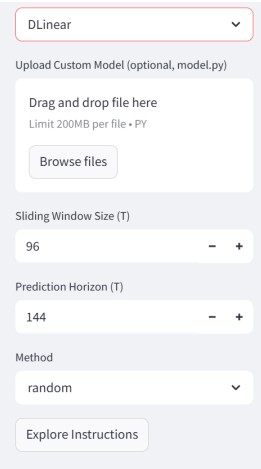

Figure 21: Configuration Options

4. **Explore Instructions**: Click on the "Explore Instructions" button. After some time, you will see the optimization process (with the successful actions over the episodes) as in Figure 22 and the reduced MSE after each episode on the validation set as in Figure 23

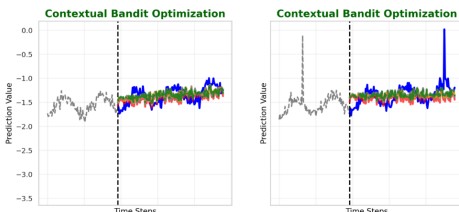

Figure 22: Successfull and failed actions during optimization

5. **Provide Feedback**: Enter your feedback in text, in any language. Be as descriptive as possible to guide the model. For example, you could say, "The amplitude of the predictions should be increased between 5% and 10% of the actual values." as in Figure 24

6. **Submit Feedback**: Click on "Submit Feedback" and then "Finalize Feedback." You will see the percentage improvement and details per channel as in Figure 25

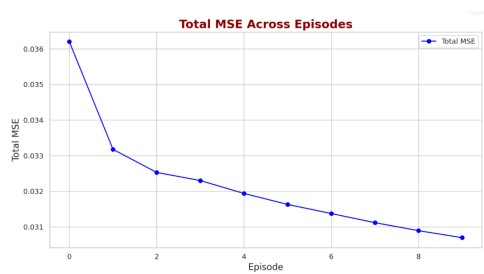

Figure 23: MSE as function of the number of episodes

**Provide Feedback**

Enter your feedback

The amplitude of the predictions should be increased between 5% and 10% of the actual values

Press Ctrl+Enter to apply

Submit Feedback

Figure 24: Example of user prompt

```
### MSE Improvement Summary

- **Total Initial MSE**: 0.0502
- **Total Final MSE**: 0.0308
- **Total Improvement**: 0.0194
- **Overall MSE Improvement**: 38.70%

**Channel-wise MSE Improvements**:

• Channel 1: 0.0188 MSE
• Channel 2: 0.0166 MSE
• Channel 3: 0.0193 MSE
• Channel 4: 0.0225 MSE
• Channel 5: 0.0196 MSE
• Channel 6: 0.0214 MSE
• Channel 7: 0.0180 MSE
```

Figure 25: Improvement Details

A.9   INSTALLATION AND USAGE FOR DEVELOPMENT

INSTALLATION INSTRUCTIONS

**Set Up the Environment**   To install the required dependencies, create and activate the conda environment:

```
conda env create -f environment.yml
```

USAGE

**Running the Code with Command-Line Arguments**   To run the post-training process and adjust the model, execute the following command:

```
  python main.py --train_path <path_to_train_data
--model <model_name> --window_size <window_size> --prediction_horizon <prediction_ho
--batch_size <batch_size> --n_samples <n_samples>
```

The available command-line arguments are as follows:

| Argument | Description | Example |
|---|---|---|
| -train_path | Path to the training data CSV file | data/train.csv |
| -model | Name of the model to use | DLinear, PatchTST, etc. |
| -model_path | (Optional) Path to a custom pre-trained model | models/custom_model.py |
| -window_size | Sliding window size for time series | 96 |
| -prediction_horizon | Prediction horizon in terms of time steps | 144 |
| -batch_size | Batch size for training | 32 |
| -n-jobs | Number of CPU for parallel computing | 1 |
| -episodes | Number of episodes for RL training | 5 |

**Running the Streamlit App**   To interact with the framework using the Streamlit interface, launch the app as follows:

```
streamlit run app_test.py
```

This will start a local server, and you can access the interface by navigating to the URL provided in the terminal.

**Workflow Overview**   The following steps outline the workflow of the post-training process:

1. **Train the Model:** Train the model using the provided training data and validate it using the validation dataset. Optionally, load a custom pre-trained model if specified.

2. **Exploration Phase:** After training, explore various actions on top of the model's predictions. These actions include adjusting amplitudes, trends, or shifting values.

3. **Human Feedback:** Provide feedback on the predictions to guide the model towards improvements. Precise feedback, such as "increase the amplitude by 5-10%", allows the model to understand the desired adjustments.

4. **Model Adaptation:** Based on the feedback, the model adapts its behavior and re-tests the adjusted predictions.

5. **Plotting Results:** The results of the model's predictions are visualized through plots, which are saved for further analysis.

**API Documentation**   The API for the framework is structured as follows:

1. app_test.py: Main script to run the Streamlit interface. Provides functionalities to explore and give feedback on model predictions.

2. `contextual_bandit.py`: Implements the contextual bandit logic for dynamically adjusting predictions based on feedback.

3. `data_extraction.py`: Contains functions for loading and preprocessing time series data.

4. `llm_interaction.py`: Functions for interacting with language models to interpret and apply human feedback.

5. `model_extraction.py`: Extracts and loads pre-trained models.

6. `plot_script.py`: Provides plotting utilities for visualizing predictions and feedback results.

### A.10 THEORETICAL MOTIVATION FOR POST TRAINING IN TIME SERIES FORECASTING

#### A.10.1 PROBLEM SETUP

Consider a supervised learning problem where we want to estimate a target variable $Y_{\text{true}}$ using a linear model. We assume that a ridge regression predictor $Y_{\text{pred}}$ has already been obtained, and we aim to improve its accuracy through an optimal affine correction of the form:

$$Y_{\text{corrected}} = aY_{\text{pred}} + b. \tag{3}$$

The goal is to determine the optimal values of $a$ and $b$ that minimize the expected mean squared error (MSE):

$$\mathcal{L}(a, b) = \mathbb{E}\left[\|Y_{\text{true}} - (aY_{\text{pred}} + b)\|^2\right]. \tag{4}$$

#### A.10.2 DERIVATION OF OPTIMAL CORRECTION PARAMETERS

Expanding the loss function:

$$\mathcal{L}(a, b) = \mathbb{E}\left[Y_{\text{true}}^2 - 2aY_{\text{true}}Y_{\text{pred}} - 2bY_{\text{true}} + a^2Y_{\text{pred}}^2 + 2abY_{\text{pred}} + b^2\right]. \tag{5}$$

**Step 1: Compute $b^*$ by setting $\frac{\partial \mathcal{L}}{\partial b} = 0$.**

$$\frac{\partial \mathcal{L}}{\partial b} = -2\mathbb{E}[Y_{\text{true}}] + 2a\mathbb{E}[Y_{\text{pred}}] + 2b. \tag{6}$$

Setting this derivative to zero and solving for $b^*$ gives:

$$b^* = \mathbb{E}[Y_{\text{true}}] - a^*\mathbb{E}[Y_{\text{pred}}]. \tag{7}$$

**Step 2: Compute $a^*$ by setting $\frac{\partial \mathcal{L}}{\partial a} = 0$.**

$$\frac{\partial \mathcal{L}}{\partial a} = -2\mathbb{E}[Y_{\text{true}}Y_{\text{pred}}] + 2a\mathbb{E}[Y_{\text{pred}}^2] + 2b\mathbb{E}[Y_{\text{pred}}]. \tag{8}$$

Substituting $b^*$ and solving for $a^*$ gives:

$$a^* = \frac{\text{Cov}(Y_{\text{true}}, Y_{\text{pred}})}{\text{Var}(Y_{\text{pred}})}. \tag{9}$$

Thus, the optimal correction parameters are:

$$a^* = \frac{\mathbb{E}[(Y_{\text{true}} - \mathbb{E}[Y_{\text{true}}])(Y_{\text{pred}} - \mathbb{E}[Y_{\text{pred}}])]}{\mathbb{E}[(Y_{\text{pred}} - \mathbb{E}[Y_{\text{pred}}])^2]}, \tag{10}$$

$$b^* = \mathbb{E}[Y_{\text{true}}] - a^*\mathbb{E}[Y_{\text{pred}}]. \tag{11}$$

#### A.10.3 THEORETICAL RISK BEFORE AND AFTER CORRECTION

**Risk Before Correction:** The mean squared error (MSE) of the original predictor is given by:

$$R_{\text{before}} = \mathbb{E}[(Y_{\text{true}} - Y_{\text{pred}})^2]. \tag{12}$$

Expanding:

$$R_{\text{before}} = \text{Var}(Y_{\text{true}}) + \text{Var}(Y_{\text{pred}}) - 2\text{Cov}(Y_{\text{true}}, Y_{\text{pred}}). \tag{13}$$

**Risk After Correction:** The mean squared error of the optimally corrected predictor is:

$$R_{\text{after}} = \mathbb{E}[(Y_{\text{true}} - Y_{\text{corrected}})^2]. \tag{14}$$

Substituting $Y_{\text{corrected}} = a^* Y_{\text{pred}} + b^*$:

$$R_{\text{after}} = \text{Var}(Y_{\text{true}}) - \frac{\text{Cov}(Y_{\text{true}}, Y_{\text{pred}})^2}{\text{Var}(Y_{\text{pred}})}. \tag{15}$$

### A.10.4 COMPARISON OF RISKS

To understand the effect of the correction, we compute the difference:

$$R_{\text{before}} - R_{\text{after}}. \tag{16}$$

Substituting the expressions:

$$R_{\text{before}} - R_{\text{after}} = [\text{Var}(Y_{\text{true}}) + \text{Var}(Y_{\text{pred}}) - 2\text{Cov}(Y_{\text{true}}, Y_{\text{pred}})]$$
$$- \left[\text{Var}(Y_{\text{true}}) - \frac{\text{Cov}(Y_{\text{true}}, Y_{\text{pred}})^2}{\text{Var}(Y_{\text{pred}})}\right]. \tag{17}$$

Simplifying:

$$R_{\text{before}} - R_{\text{after}} = \text{Var}(Y_{\text{pred}}) - 2\text{Cov}(Y_{\text{true}}, Y_{\text{pred}}) + \frac{\text{Cov}(Y_{\text{true}}, Y_{\text{pred}})^2}{\text{Var}(Y_{\text{pred}})}. \tag{18}$$

Rewriting using the identity:

$$\left(x - \frac{a}{x}\right)^2 \geq 0 \quad \text{for all } x > 0, \tag{19}$$

by setting $x = \text{Var}(Y_{\text{pred}})$ and $a = \text{Cov}(Y_{\text{true}}, Y_{\text{pred}})^2$, we obtain:

$$R_{\text{before}} - R_{\text{after}} = \left(\sqrt{\text{Var}(Y_{\text{pred}})} - \frac{\text{Cov}(Y_{\text{true}}, Y_{\text{pred}})}{\sqrt{\text{Var}(Y_{\text{pred}})}}\right)^2. \tag{20}$$

Since the square of any real number is always non-negative:

$$R_{\text{before}} - R_{\text{after}} \geq 0. \tag{21}$$

### A.10.5 CONCLUSION

This derivation shows that the correction always reduces the risk (or at worst, leaves it unchanged). The correction is most effective when $Y_{\text{pred}}$ is correlated with $Y_{\text{true}}$, and it does not increase the error in any case. This result shows that the correction always reduces the mean squared error.

### A.11 PROOF OF THE *Upper Bound on the Risk of the Corrected Prediction* THEOREM

For completeness we give here the general theorem (for $K > 2$) and its assumptions.

**Assumption 1 (Gaussian Squared-Error Model)** *For all indices $k$, hyperparameters $\beta$, and the base model $f_\theta$, we assume that for $(X, Y) \sim \mathcal{D}$,*

$$\left((g_{k,\beta} \circ f_\theta)(X) - Y\right)^2 \quad \textit{follows a Gaussian distribution with mean } R(g_{k,\beta} \circ f_\theta) \textit{ and variance } \sigma^2 > 0.$$

*This provides a convenient concentration model for the empirical risk estimates used by the algorithm.*

**Theorem 2 (Upper Bound on the Risk of the Corrected Prediction)** *Let $f_\theta$ be a base predictor, $(g_{k,\beta^*})_{k=1}^K$ a set of corrective actions, and assume a total evaluation budget $T$ under $M = 1$. Under Assumption 1, applying Successive Halving to select a correction yields:*

$$\mathbb{E}\left[R(g_{k_T, \beta^*} \circ f_\theta)\right] \leq \frac{1}{\nu} \sum_{k=1}^K \bar{R}(k) \prod_{r=0}^{\log_2 K - 1} \left[(k-1)\,\Phi\!\left(-\Delta_{\min,k}^+ \tau_{\text{dec}}(r)\right) + (K-k)\,\Phi\!\left(-\Delta_{\min,k}^- \tau_{\text{inc}}(r)\right)\right],$$

*where*

$$\bar{R}(k) := \min\{R(f_\theta), R(g_{k,\beta^*} \circ f_\theta)\}, \quad \nu := \frac{K^{\log_2 K}}{2\sqrt{2}^{\log_2 K(\log_2 K+1)}},$$

$$\tau_{\text{dec}}(r) := \sqrt{\frac{T2^r}{2\sigma^2 K \log_2 K} - \frac{1}{2\sigma^2}}, \quad \tau_{\text{inc}}(r) := \sqrt{\frac{T2^r}{2\sigma^2(K \log_2 K - 2^r \log_2 K)}},$$

$$\Delta^+_{\min,k} := \min_{\substack{j\neq k \\ R(k)>R(j)}} \big(R(k) - R(j)\big), \quad \Delta^-_{\min,k} := \min_{\substack{j\neq k \\ R(k)<R(j)}} \big(R(k) - R(j)\big).$$

**Proof 1 (Sketch of the proof)** *We provide here a self-contained outline; the full derivation follows the style of* Karnin et al. *(2013) for Successive Halving.*

**Step 1: Decomposition.** *Each corrective action $g_{k,\beta^*}$ has a fixed true risk $R(g_{k,\beta^*} \circ f_\theta)$. Only the index $k_T$ selected by the algorithm is random because it depends on noisy empirical risk estimates. Hence,*

$$\mathbb{E}\big[R(g_{k_T,\beta^*} \circ f_\theta)\big] = \sum_{k=1}^K \min\big(R(f_\theta), R(g_{k,\beta^*} \circ f_\theta)\big) \, \mathbb{P}[k_T = k].$$

**Step 2: Bounding the selection probability.** *Successive Halving proceeds over $n_r = \log_2 K$ rounds. At round $r$, each remaining action in the set $S_r$ is evaluated $t_r$ times and half of them are discarded. For a fixed $k$, the number of competing actions with lower empirical risk than $k$ can be expressed as a sum of Bernoulli variables whose expectations are Gaussian tail probabilities:*

$$\mathbb{E}[N_{r,k}] = \sum_{j\neq k} \Phi\Big(-\frac{\Delta_{k,j}\sqrt{t_r}}{\sqrt{2}\sigma}\Big).$$

*Using Markov's inequality and introducing the smallest positive and negative risk gaps $\Delta^+_{\min,k}$ and $\Delta^-_{\min,k}$, we obtain a product-form upper bound on $\mathbb{P}[k_T = k]$ across all rounds.*

**Step 3: Plug into the risk expression.** *Substituting this bound on $\mathbb{P}[k_T = k]$ into the decomposition above yields the stated inequality.*

**Discussion.** This theorem formalizes how Successive Halving selects the best correction under limited evaluation budget. The bound shows that the expected risk converges to the risk of the optimal correction at an exponentially fast rate, with the convergence speed determined by the risk gaps $\Delta^\pm_{\min,k}$ and the budget $T$. In particular, larger gaps between actions accelerate the identification of the optimal correction—exactly mirroring the empirical behavior observed in our experiments.

