# OpenReview forum: "Human-in-the-Loop Adaptive Optimization for Improved Time Series Forecasting"
_ICLR.cc/2026/Conference — Submitted to ICLR 2026_

### Official Review · Reviewer_pnJN · 2025-10-29

**Soundness:** 3
**Presentation:** 2
**Contribution:** 2
**Rating:** 4
**Confidence:** 3

**Summary:**

The paper presents a novel post-training adaptive correction framework designed to enhance the accuracy of time series forecasting models without retraining or architectural modification. A key innovation is the integration of a Human-in-the-Loop component, enabling domain experts to guide corrections through natural language feedback. These human instructions are parsed by a large language model into executable transformation actions that are then validated and optimized within the same adaptive pipeline. The paper provides both theoretical foundations (affine correction guarantees lower MSE, Theorem 1) and empirical analyses across multiple benchmarks, demonstrating consistent accuracy improvements with minimal overhead.

**Strengths:**

1. The framework is fully model-agnostic, compatible with any forecasting architecture without altering parameters or requiring retraining. This plug-and-play property makes it directly usable in production systems.

2. HITL component represents a novel integration of human expertise via natural language, supported by a safe validation mechanism that prevents code injection and ensures feedback quality (Algorithm 2, Fig. 5). This bridges the gap between machine optimization and domain expert intuition, offering genuine practical utility.

**Weaknesses:**

1. The paper claims low computational overhead, yet the runtime analysis (Table 3) only compares optimization time to base model training cost for a few horizons on a single dataset (ETTh1). There is no evaluation on larger models or streaming scenarios where real-time adaptation is claimed to be possible. Moreover, the runtime excludes the LLM inference and validation costs required for parsing human feedback, which could be nontrivial in practice.

2. Safety and robustness of LLM-based feedback translation are insufficiently validated. Algorithm 2 specifies a validation routine, but the paper never clarifies the criteria for safety or correctness. There is no discussion of how the system handles ambiguous, adversarial, or syntactically valid but semantically incorrect feedback. This omission raises concerns about reliability in real deployment.

3. The role and quantifiable contribution of the Human-in-the-Loop component are inadequately analyzed. The authors present qualitative examples but provide no quantitative ablation isolating HITL performance relative to the automated optimization baseline. It is therefore unclear how much of the reported gains are due to algorithmic improvements versus human intervention. Without such evidence, the practical value of the human feedback module remains speculative

**Questions:**

1. Could the authors include an ablation comparing (i) automated optimization only, (ii) HITL without optimization (manual corrections only), and (iii) full system performance, to quantify human feedback benefits?

2. What is the end-to-end computational cost, including LLM feedback parsing?

3. Is there a theoretical guarantee that RL- or GA-based transformation policies converge or do not degrade performance under noisy validation data?

---

> ### Author Response · Authors · 2025-11-19
> **Response to reviewer pnJN**
>
> We would like to thank the reviewer for their positive and constructive comments. Below, we address your concerns and provide detailed answers to your questions.
>
> **Computational Overhead and Runtime Analysis**
>
> We appreciate the reviewer’s insightful comments regarding the runtime analysis. To clarify, the runtime in Table 3 is focused on the post-training optimization process, which operates independently of the base model’s training cost. Whether the base model has 10 parameters or 1 billion, the post-training runtime remains consistent because it only operates on the forecasted values, not on the model parameters. The primary factors influencing the post-training computational overhead are the number of actions and the horizon length, which is why these two parameters are used for comparison in the table.
>
> Regarding the LLM inference and validation costs for parsing human feedback, we agree that these components can introduce additional overhead. However, it is important to note that these are optional and do not impact the core contribution of the paper, which is the post-training optimization process. The LLM inference time is typically very fast (on the order of a few seconds for large models), and the validation step depends on the complexity and length of the feedback, which varies from user to user. Therefore, benchmarking these costs is challenging and outside the scope of the paper, as they depend on the specific user interaction and environment.
>
>
> **Safety and Robustness of LLM-Based Feedback Translation**
>
> We appreciate the reviewer’s concern regarding the safety and robustness of LLM-based feedback. For ambiguous or semantically incorrect feedback, our optimization process (using reinforcement learning or bandit methods) minimizes the impact by focusing solely on objective performance improvements. We address this issue through experiments (Figures 16, 17, and 18) that demonstrate how the system handles ambiguous prompts. These figures show cases where the human-in-the-loop component either improves performance (Figures 13, 14, and 15) or results in no improvement, as the system detects inconsistencies and disregards ineffective feedback.
> By focusing on performance metrics, our approach ensures that the system remains reliable even with less precise or inconsistent human input (see also the general comments to all reviewer for further details).
>
> **Human-in-the-Loop Contribution and Ablation Analysis:**
>
> In the main paper, we have isolated the influence of the Human-in-the-Loop component, which is optional, to evaluate its impact alongside the automated optimization. Specifically, Table 1 and Table 2 show results obtained purely through the automated framework, with optimization over a fixed set of actions, serving as a baseline ablation of the paper's algorithmic improvements. In contrast, Figures 13, 14, and 15 illustrate successful cases where HITL feedback improves performance, with the third subplot showing the quantitative improvement due to the human prompt, isolating it from the automated process. Additionally, we have included three new cases (Figures 16, 17, and 18) where ambiguous prompts led to no performance gain, demonstrating that ineffective feedback is recognized and discarded. These ablation studies, including the quantitative analysis in the third subplot of each figure, provide a clear and independent evaluation of both components, highlighting the practical value of HITL feedback while confirming the reproducibility of algorithmic improvements.
>
> **Theoretical Guarantees on RL/GA-Based Transformation Policies**
>
> We appreciate the reviewer’s insightful question. We do provide a theoretical guarantee for the bandit algorithm used in our optimization process, ensuring convergence and stability under noisy validation data. However, we acknowledge that providing a similar theoretical guarantee for the reinforcement learning (RL) and genetic algorithms (GA) used for transformation policies would be a valuable addition for future work. Currently, the RL and GA approaches are empirically validated and have shown robustness in practice, but we agree that formal guarantees would strengthen the approach further. We will note this limitation in the revised manuscript and highlight the potential for future theoretical analysis in these areas.
>
> **Additional comments**
> If you have any additional questions about the paper, we would be happy to provide further clarification. Please let us know if you need any additional information, clarification or modification that we could provide for you to improve your score.

---

> > ### Comment · Reviewer_f45E · 2025-11-26
> >
> > I would like to thank the authors for their thoughtful and detailed response. However, after careful consideration, I remain unconvinced by the novelty and technical contribution of the proposed approach overall. Therefore, I will maintain my initial score.

---

### Official Review · Reviewer_Fn8E · 2025-10-29

**Soundness:** 2
**Presentation:** 1
**Contribution:** 2
**Rating:** 4
**Confidence:** 2

**Summary:**

This paper presents a model-agnostic framework with two components:
- A post-training affine transformation to refine predictions that employs existing search strategies: random search, bandits, genetic algorithm, and reinforcement learning to optimizes parameters of the transformation and covers four actions: Scale Amplitude, Piecewise Scaling, Linear Trend, and Min/Max Adjustment.
- A human in the loop mechanism that provides natural language instructions such as “Increase values above the 80th percentile” that translate this to an action that can be added to the action pool

The utility of the proposed method was studied on standard benchmarks and authors showed performance improvement in terms of MSE percentage change across various time series forecasting models.

**Strengths:**

- The paper provides some theoretical insights which can be interesting for the community.
- I am happy to see that authors conducted experiments beyond the usual time series forecasting benchmarks.

**Weaknesses:**

1- I am not sure if I fully agree with the role of the human in the loop here. For example, in figure 5(a) increasing the amplitude of predictions by x% it could and should be something that the objective function does as to me it is not really a domain knowledge. Is it really necessary that a human provide this type of feedback? I don’t believe so. In addition, code generation for a new action via prompting is also not something novel. Therefore, overall I am struggling to understand the contribution here of Human-In-The-Loop Feedback part.

2- Although authors have provided time analysis based on different actions however, performance analysis for different number of actions is missing.

3- Overall, I had a hard time following the paper as some details which may seem trivial for authors are missing and I think presentation can be significantly improved. For example:
- Figure 3 is not cited anywhere in the paper, what is the significance of it? also, is the regularization in this figure for the ridge regression?
- In line 213, what’s the definition of the loss? Is this the same loss as in line 1588?
- What are the rows in the table provided in section A.3.2?
- In line 242, what does “and consistency is checked on the training set to ensure generalizable gains” mean? What’s the definition of consistency here? gap between train and validation loss?
- I think the paper will benefit from more in-depth elaboration around post-training and inference. For example, what exactly happens at post-training meaning are the parameters being updated sequentially or batch-wise?

I’d be happy to revisit my score if authors address these weaknesses.

**Questions:**

What is the applicability domain of this framework as a whole? for example, looking at table 6 it seems most improvements are marginal with the exception of two transformer-based model. Does this mean the presented framework is more suitable for transformers? If so, why? These discussions are missing in the paper specially based on the model-agnostic claim of the paper.

---

> ### Author Response · Authors · 2025-11-19
> **Response to reviewer Fn8E**
>
> We thank the reviewer for their positive and constructive feedback. Below, we address the key concerns and provide detailed answers.
>
> **Role of Human-in-the-Loop Feedback**
>
> While the LLM code generation itself is not novel, it accelerates the process of incorporating human expertise into the optimization framework. The core contribution of our work lies in the automated optimization over a fixed set of actions (Tables 1 and 2), with the human-in-the-loop component acting as an optional enhancement. Base forecasting models like PatchTST may lack domain-specific knowledge, such as recognizing that negative forecasts are infeasible, that predictions should fall within certain ranges, or that some phenomena, like voltage saturation, are not linear. These constraints are difficult to incorporate directly into the base model without risking overfitting or introducing unsafe, time-consuming manual adjustments. The HITL framework allows users to inject valuable domain knowledge in a safe, interpretable, and efficient way, which is then optimized on a validation set with overfitting checks. This enables simple yet effective improvements without modifying the base model, and we believe this is the key practical value of HITL feedback.
>
> **Performance Analysis for Different Numbers of Actions**
>
> We have added an analysis in Table 8 showing how increasing the number of actions improves performance, addressing the reviewer’s concern.
>
> **Presentation and Missing Details**
>
> We apologize for the missing details and will be more than happy to update and clarify remaining reviewers concern.
>
> Figure 3: We add a citation and a clear explanation of Figure 3 to clarify its significance within the context of the paper. Indeed the figure shows the MSE as function of the ridge regularization both for the initial ridge and for the post training layer applied on ridge where we can verify what is claimed in Theorem 1, an improvement for any regularization parameter. We add this explanation in the new updated version after Theorem 1.
>
> Loss Definition (Line 213 vs Line 1588): We clarify the loss function definition can be any loss defined by the user (by default we use MSE and give it as example in the algorithm). The loss in line 213 refers to the loss used during the post training optimization, and line 1588 refers to the loss function of the ridge regression optimization. We use in the new version a different notation for the two losses.
>
> Rows in Table A.3.2: We provide a more detailed description of the rows in Table A.3.2, explaining what each row represents and its role in the broader analysis. The different rows for each dataset are different horizon lengths for the forecasting.
>
> Consistency (Line 242): We clarify our definition of consistency in the updated version of the paper. Specifically, consistency refers to whether the selected optimal set of actions actually improves performance on the training samples. In other words, we evaluate the chosen set of actions on the training data and check if applying these actions reduces the training loss compared to its original value before post-training.
>
> Post-Training and Inference: We are not entirely sure we understand the reviewer’s question. After training, the post-training layer is learned, and the optimal set of actions is saved for use during inference. This allows us to compose the initial prediction with the optimized actions. For example, if we train PatchTST to obtain $f_\Theta^\star$, we then perform post-training to learn an optimized set of actions $g^\star$. During inference, the final prediction is obtained by combining both as $f_\Theta^\star \circ g^\star$
>
> **Applicability Domain and Model-Agnostic Claim**
>
> Our framework can be applied to any forecasting task, with the automated layer improving performance through simple, interpretable corrections. The human-in-the-loop component is useful when an expert identifies potential errors (e.g., physical domain knowledge, physical constrains or visual artefacts from observing several forecasts). The expert's feedback is evaluated through optimization to ensure it improves performance. The user provides suggestions, and the optimization validates their effectiveness.
>
> Regarding the model-agnostic claim, we observe a notable performance gain across all base models. We agree that some transformer-based approaches exhibit larger improvements; however, even methods such as SegRNN, DLinear, and PatchTST show average improvements of 18.58%, 14.71%, and 12.28%, respectively. The highest improvement overall is achieved with DLinear, indicating that our approach enhances performance regardless of the underlying model. As future work, it would be interesting to investigate and predict in advance which models are most likely to benefit from this approach.
>
> **Additional Comments**
>
> If you have further questions or need additional clarification, please let us know. We are happy to provide more information to improve your review.

---

> > ### Comment · Reviewer_Fn8E · 2025-11-26
> > **Thank you**
> >
> > I appreciate authors' efforts and response to comments. Most of my concerns have been addressed and I gladly increase my score. The only open issue to me is whether the level of contribution is sufficient for ICLR.

---

### Official Review · Reviewer_f45E · 2025-11-01

**Soundness:** 3
**Presentation:** 2
**Contribution:** 2
**Rating:** 4
**Confidence:** 5

**Summary:**

This paper proposes an adaptive optimization framework for time series forecasting, applied after training. Rather than retraining base predictors, the method uses a set of interpretable output transformations to correct systematic prediction errors. The approach can incorporate Human-In-The-Loop (HITL) guidance, in which domain experts provide natural-language suggestions that are converted into executable actions by an LLM. The authors provide a theoretical analysis demonstrating that affine corrections can reduce MSE under matched distributions, and offer a budget-dependent performance bound for SH-HPO. Experiments conducted across multiple datasets and base predictors demonstrate consistent improvements in MSE and cross-metric performance, with relatively low computational overhead.

**Strengths:**

1. The paper addresses the highly practical and impactful problem of improving time-series forecasts without having to retrain base models.
2. The authors present a transparent and comprehensible design for the transformation space.
3. The paper offers theoretical support for the proposed approach, including proof that affine corrections can reduce mean squared error (MSE) under aligned distributions, and performance bounds for Successive-Halving-based optimization.

**Weaknesses:**

1. The method relies heavily on the assumption that the validation and test distributions are aligned. However, the paper does not sufficiently analyze how the approach behaves under distribution drift, which is common in real-world time series environments.
2. The Human-in-the-Loop pipeline lacks detailed safeguards to prevent information leakage, bias, or iterative overfitting to the validation set.
3. The experimental analysis does not adequately address failure cases, particularly instances where the proposed corrections result in a decline in performance, as observed in PatchTST.

**Questions:**

1. The paper claims that LLMs are used only for language editing, yet the role of the LLM in generating executable code contradicts this. Furthermore, the authors do not provide ablation studies or prompt details to clarify how LLM behavior affects the system’s stability or reproducibility.
2. Could the authors please explain their strategy for handling multivariate target series? This should include details of whether transformations are applied independently or jointly, and how cross-dimensional constraints or correlations are preserved.

---

> ### Author Response · Authors · 2025-11-19
> **Response to reviewer f45E**
>
> We would like to thank the reviewer for their positive and constructive comments. Below, we address your concerns and provide detailed answers to your questions
>
> ### **Distribution Drift and Performance Under Shifting Distributions**
> We fully acknowledge the concern that distribution drift is common in real-world time series forecasting, and the potential impact on model performance is a valid consideration. Our method is designed with this in mind and, as mentioned in the common answer to all reviewers, stress tests and empirical evaluations have been conducted on datasets that exhibit distribution shifts. For example, we tested our framework on several benchmarks known to exhibit distribution drift (e.g., ETTh1, NASDAQ among others in our 12 datasets), where our method consistently outperforms the base model, even under such shifts. This suggests that the approach is robust to potential distribution mismatches between validation and test sets.
>
> ### **Human-in-the-Loop Safeguards: Information Leakage and Overfitting**
> We understand concerns about overfitting, information leakage, and bias. To prevent overfitting, we perform optimization using reinforcement learning (bandit) over the pool of actions generated by the LLM on the validation set. The "check_overfitting" function then ensures that the optimal actions found improve performance on the training set as well. This cross-check helps confirm genuine generalization.
>
> Information leakage is prevented since the LLM only sees the user's prompt, not model outputs or raw data. The LLM functions purely as a code generator, and since the task is objective (with clear template format for the code), we do not expect significant bias.  The LLM’s role is strictly as a code generator, not a reasoner, and since the task is objective (converting prompts to a template code), we don't expect significant bias. A data scientist could perform the same task, with the LLM simply speeding up and automating the process.
>
> ### **Failure Cases and Performance Decline**
> The performance degradation with PatchTST is rare. Out of the 12 datasets tested, only 1 showed moderate degradation (-2.25%), while the average improvement across all datasets is +14.41%, with a peak improvement of 51.26%. In Table 1, where we compare optimization strategies, we focus on ETTh1, the only dataset with observed degradation, but this is an exception. Table 2 highlights that performance degradation is generally uncommon.
>
> ### **No LLM was use to conduct research**
> There might be a misunderstanding: we did not use an LLM to conduct the research underlying this work; it was only used to help polish the writing. Although our method leverages an LLM as a component (to allow a safe human-in-the-loop option), the core ideas and contributions were conceived and developed by the authors, not generated by an LLM.
>
> ### **On prompt details**
> In response to the concern about ablation studies and prompt details, we provide qualitative case studies in Figures 13, 14, and 15 (on one side) and Figures 16, 17, and 18 (on the other). On the left of each figure, we display the user's prompt (in the title) and the corresponding time series. In the middle, we show the code generated by the LLM, and on the right, we illustrate the performance improvement after optimization and overfitting check. Specifically, the first three figures demonstrate improvements driven by meaningful prompts, while the last three show ambiguous or nonsensical prompts that were discarded by the system, resulting in no performance degradation. This highlights how the system maintains stability and reproducibility, even when some prompts do not provide useful feedback.
>
> ### **Handling Multivariate Target Series**
>
> Regarding the handling of multivariate target series, we clarify that, for the current version of our method, multivariate time series are handled independently. Specifically, we apply transformations to each target series individually. This choice was made due to the complexity of joint modeling, which we are exploring for future work.
>
> ### **Additional comments**
> If you have any additional questions about the paper, we would be happy to provide further clarification. Please let us know if you need any additional information, clarification or modification that we could provide for you to improve your score.

---

### Official Review · Reviewer_z96C · 2025-11-01

**Soundness:** 3
**Presentation:** 3
**Contribution:** 3
**Rating:** 8
**Confidence:** 2

**Summary:**

The paper proposes a lightweight, model-agnostic post-training adaptive optimization layer that refines forecasts by selecting and tuning a small set of interpretable output transformations via various algorithms, and optionally accepts natural-language expert guidance translated to executable candidate actions by an LLM. The authors theoretically justify affine corrections, describe a discrete-action + continuous-parameter action space, and show empirical improvements across many forecasting benchmarks, reporting consistent MSE reductions and a web demo.

**Strengths:**

1. The paper presents a simple yet practical idea, post-training output transformations, that can be applied to any base forecasting model without retraining and therefore has immediate engineering value.
2. The theoretical statement that an optimal affine correction cannot increase MSE is clean and appropriately motivates richer post-training actions, and the action design is interpretable, compact, and easy for practitioners to reason about and extend. Comparing several search strategies (random search, SH-HPO bandits, PPO, GA) and reporting their relative merits gives the reader actionable guidance about tradeoffs between simplicity, runtime, and performance.
3. Integrating human feedback via natural language, with an LLM translating prompts into candidate actions which are then validated before use, is a useful and modern design that can make forecasting pipelines more usable to domain experts.
4. The paper includes reproducibility and ethics statements and points to appendices and an interactive demo, which increases transparency and practical verifiability.

**Weaknesses:**

1. The claimed average improvements are modest and inconsistent across model/dataset pairs, and several cases show small or negative gains, so it is not yet clear when practitioners should expect reliable improvements versus when the method risks degrading performance.
2. The method depends critically on a representative validation set and the choice of validation budget. When validation and test distributions mismatch, such as concept drift, the optimization could overfit to validation artifacts and harm generalization. The paper states this issue, but provides limited empirical stress tests under realistic distribution shifts.
3. he human-in-the-loop path converts natural language to executable code. This opens safety and security questions (sandboxing, injection, resource use) that are only briefly sketched. The paper asserts validation and that the LLM never accesses raw data, but concrete safeguards, adversarial scenarios, and human-study evidence are missing.

**Questions:**

1. Please explain how validation sets were constructed and whether the optimization budget (number of evaluations) was the same for all datasets and horizons; how sensitive are results to this budget?
2. Please explain why some model/dataset combinations (e.g., PatchTST on some sets) show degraded performance after optimization, and whether action constraints or regularization can avoid such negative cases.

---

> ### Author Response · Authors · 2025-11-19
> **Response to reviewer z96C**
>
> We would like to thank the reviewer for their positive and constructive comments. Below, we address your concerns and provide detailed answers to your questions
>
> ### **Consistent Improvements Across Datasets**
> Our method achieves an average improvement of over 14% across all datasets (see Table 2), with the highest improvement of 57.85% across 12 competitive datasets and 7 state-of-the-art time series forecasting models. Although a 14% average error reduction may sound modest, it is sizable in the current long-term time series forecasting literature.
>
> Our gains are computed against strong modern baselines in a unified pipeline and come from a generic post-training correction that can be applied on top of existing models, so they add to the improvements brought by the base architectures.
>
> ### **Rare Cases of Negative Performance**
> Out of 84 individual model-dataset pairs, we observe only 4 cases of negative performance, with an average degradation of -0.95% and a maximum degradation of -2.25%. These rare negative cases can be explained both by deviations between the validation and test sets (see the major concerns) and by the fact that our post-training action search improves performance only with high probability.
>
> ### **Theoretical Justification for Robustness**
> The occasional minor degradation in performance can be attributed to the method’s deliberate design. By using a simple, constrained set of interpretable actions, we introduce controlled bias to reduce variance and mitigate overfitting, rather than aiming for a significant reduction in the base model’s bias (with complex uninterpretable actions). This approach stabilizes the model, as shown by the minimal performance degradation. Additionally, to prevent overfitting, we validate that the optimized action pool improves performance on both the training and validation sets. Any performance degradation serves as a signal of overfitting, supporting the effectiveness of our design. In essence, our focus is on learning the residuals from the base model while managing the bias-variance tradeoff, ensuring low variance without overly complicating the model and letting it to be interpretable.
>
> ### **Further Evidence of Robustness**
> In the Appendix (Figure 9, page 24) and in Figures 11–12 of the revised version, we report train, validation, and test losses during optimization, showing that limiting model complexity systematically reduces overfitting and supports the robustness of our approach. Overall, degradations are rare and remain within expected bounds. We update the paper to briefly summarize these results, discuss the few negative improvements, and provide an intuitive explanation of why the method remains robust.
>
> ### **Degraded Performance in Some Model/Dataset Combinations**
> The performance degradation with PatchTST is rare. Out of the 12 datasets tested, only 1 showed moderate degradation (-2.25%), while the average improvement across all datasets is +14.41% for PatchTST, with a peak improvement of 51.26%. In Table 1, where we compare optimization strategies, we focus on ETTh1, the only dataset with observed degradation, but this is an exception. Table 2 highlights that performance degradation is generally uncommon.
>
> ### **Further discussions**
> The optimization budget was set to 10 evaluations across all datasets. Our theoretical analysis emphasizes the significance of this budget (see Corollary 2 page 6). While increasing the budget would likely yield even better results (as stated by the theoretical result), our experiments across multiple datasets showed that 10 evaluations were sufficient to achieve satisfactory performance. This demonstrates that the method is efficient, providing strong results even with a relatively small optimization budget.
>
>
> ### **Additional comments**
> If you have any additional questions about the paper, we would be happy to provide further clarification. Please let us know if you need any additional information, clarification or modification that we could provide for you to improve your score.

---

### Author Response · Authors · 2025-11-19
**Major concerns and updates**

We thank the reviewers for acknowledging our model-agnostic approach, which enhances time-series forecasting without retraining and provides solid theoretical insights (Reviewer z96C, Reviewer f45E). We appreciate the recognition of our transparent design, actionable search strategy guidance, and the integration of human feedback to improve usability and safety (Reviewer z96C, Reviewer pnJN). We also value the positive feedback on our experiments beyond standard benchmarks (Reviewer Fn8E). Below, we address the main concerns and summarize the changes made to the paper (in blue in the updated version), followed by specific responses to each reviewer.

### **About the generalization of our post-training improvement (z96C, f45E)**

We follow standard practice for validation in time series forecasting. For benchmarks with predefined validation splits (e.g., ETTh1, ETTm1, ETTh2, ETTm2), we use the provided val segments; for others, we create a chronological split, reserving the last 30% of the training data for validation. As in prior work (e.g., PatchTST, DLinear), this validation set is used for model selection / early stopping, and we apply exactly the same split for our method; all reported gains are computed after this selection step.

Although distribution shifts between validation and test can affect generalization, our experiments on benchmarks with pronounced shifts (e.g., ETTh1, NASDAQ) show that our approach consistently improves over the base model, suggesting robustness to validation-specific artifacts. Additional experiments varying the training/validation ratio indicate that performance (quantitative MSE reduction) is stable across different split sizes, and that our gains are not highly sensitive to the exact partition (see updated Figure 8, page 20).

### **About safety with regards to the Human-Feedback (z96C, f45E, pnJN)**
In our design, the user provides only a high-level prompt (e.g., “increase prediction amplitude by a factor between 5% and 30% of the initial prediction”), which the LLM maps to a function following a strict template (see page 27 of the updated version) parameters that can be optimized. Also note that the Python interpreter runs as a non-root user, preventing access to system or other sensitive files via OS-level permissions.

Then each proposed action is optimized using one of the four methods described in the paper on both training and validation sets and added to the action pool only if it yields a measurable improvement, already limiting harmful or misguided inputs.

We further introduce an optional "safe mode", in which LLM-generated code is shown to the user for inspection and scanned with the malicious-source-code detector of Tsfaty and Fire (2023). This adds only minor overhead, since each action is checked once at creation time and code snippets are short.

To assess robustness, we also report some cases with ambiguous or nonsensical prompts (Figure 16, 17, 18 of the updated version).
### **Summary of updates**

We include a more detailed discussion in the paper regarding the safety and security measures of the human-in-the-loop framework, including the stress tests and adversarial scenarios that demonstrate its robustness. Additionally, we provide further clarification on the construction of validation sets, robustness to size of the validation set, and the potential impact of number of actions on the results. In the following, we detail our course of action on specific points:

- Validation Set Analysis: We analyze the impact of different train/validation split ratios on the overall framework performance. The improvements (MSE reductions compared to based model as defined in Equation 1) for various ratios are shown in Figure 8 page 20.

- Overfitting and Validation Set Discussion: To further explore overfitting, we provide in Figure 11 and 12, the training, validation, and test loss landscapes across optimization episodes (number of steps performed on the validation set). This is done for several datasets and includes multiple metrics, demonstrating consistent agreement between the loss curves. This reinforces the low risk of overfitting, which is due to the simple, interpretable set of actions chosen.

- Robustness Analysis: In Figure 16, 17, 18 (page 30), we analyze the robustness of the approach with respect to different prompts of the LLM, highlighting the stability of the method, aided by the safeguards (optimization over the user-defined action set).

- Summarize Table 2 to highlight the 14% overall improvement (with a peak of 57.85%) across 12 datasets and 7 models, noting the rare performance degradation (4 cases out of 84), with an average decline of -0.94% and a maximum of -2.25%.

- Clarify the validation set choice: standard validation benchmarks for some datasets, and a standard training/validation split for others.

**References**

Tsfaty and Fire (2023) Malicious source code detection using a translation model

---

### Meta-Review · Area_Chair_5sBq · 2026-01-06

**Summary:**

Reviewers generally agreed that the paper addresses a practical and relevant problem—improving time-series forecasting via a model-agnostic post-training correction layer—and appreciated the breadth of experiments and the clean theoretical result for affine correction under MSE. However, the discussion consistently raised concerns about whether the work meets the novelty and conceptual contribution bar expected at ICLR. In particular, several reviewers viewed the core methodology as closely related to existing post-hoc calibration, residual correction, or ensemble-style adjustment techniques, with the main contribution lying more in system integration and empirical evaluation than in new algorithmic insight.

Additional concerns focused on the reliance on a representative validation set, limited stress-testing under realistic distribution shift, and the unclear marginal benefit of the human-in-the-loop component, which remains largely qualitative and insufficiently isolated from the automated optimization baseline.

**Reviewer Concerns:**

The rebuttal addressed several reviewer concerns by clarifying the validation protocol, adding robustness analyses, and providing more detail on safety mechanisms and failure cases in the human-in-the-loop pipeline. These updates improved confidence in the empirical results and mitigated some overfitting and security concerns.

However, core concerns remain outstanding. In particular, reviewers remain unconvinced about the novelty of the overall contribution relative to existing post-hoc calibration and correction methods, and the incremental value of the human-in-the-loop component is not quantitatively isolated through clear ablations. Additionally, robustness under distribution shift and realistic deployment scenarios is still only partially explored. As a result, while the rebuttal strengthened the paper, it did not fully resolve the main concerns affecting the final decision.

**Reviewer Scores:**

Reviewers who were initially positive or borderline-positive would likely maintain their scores or increase them slightly, as several practical and safety-related concerns were addressed in the rebuttal and additional experiments. In contrast, reviewers who expressed reservations about novelty and technical contribution indicated in follow-up comments that these issues remained unresolved and would therefore keep their original ratings. Overall, the score distribution would remain mixed and centered around the borderline, without clear convergence toward acceptance.

---

### Decision · Program_Chairs · 2026-01-26

Reject